# Surface water and groundwater: Unifying conceptualization and quantification of the two "water worlds"

Brian Berkowitz[1], Erwin Zehe[2]

1. Department of Earth and Planetary Sciences, Weizmann Institute of Science, Rehovot 7610001, Israel
2. Karlsruhe Institute of Technology (KIT), Karlsruhe, Germany

*Correspondence to:* Brian Berkowitz (brian.berkowitz@weizmann.ac.il)

**Abstract.**

While both surface water and groundwater hydrological systems exhibit structural, hydraulic and chemical heterogeneity, and signatures of self-organisation, modelling approaches between these two "water world" communities generally remain separate and distinct. To begin to unify these water worlds, we recognize that preferential flows, in a general sense, are a manifestation of self-organisation; they hinder perfect mixing within a system, due to a more "energy efficient" and hence faster throughput of water and matter. We develop this general notion by detailing the role of preferential flow for residence times and chemical transport, as well as for energy conversions and energy dissipation associated with flows of water and mass. Our principal focus is on the role of heterogeneity and preferential flow and transport of water and chemical species. We propose, essentially, that related conceptualizations and quantitative characterisations can be unified in terms of a theory that connects these two water worlds in a dynamic framework. We discuss key features of fluid flow and chemical transport dynamics in these two systems – surface water and groundwater – and then focus on chemical transport, merging treatment of many of these dynamics in a proposed quantitative framework. We then discuss aspects of a unified treatment of surface water and groundwater systems in terms of energy and mass flows, and close with a reflection on complementary manifestations of self-organisation in spatial patterns and temporal dynamic behaviour.

**Keywords:** Chemical transport, Continuous Time Random Walk (CTRW), self-organisation, preferential flow

# 1 INTRODUCTION

While surface and subsurface flow and transport of water and chemicals are strongly interrelated, the catchment hydrology ("surface water") and groundwater communities are split into two "water worlds". The communities even separate terminology, writing "surface water" as two words but "groundwater" as one word!

At a very general level, it is well recognized that both catchment systems and groundwater systems exhibit enormous structural and functional heterogeneity, which are, e.g., manifested through the emergence of preferential flow and space-time distributions of water, chemicals, sediments and colloids, and energy across all scales and within/across compartments (soil, aquifers, surface rills and river networks, full catchment systems, and vegetation). Dooge (1986) was among the first hydrologists who distinguished between different types of heterogeneity – namely, between stochastic and organised/structured variability – and reflected upon how these forms affect predictability of hydrological dynamics. He concluded that most hydrological systems fall into Weinberg's (1975) category of organised complexity – meaning that they are too heterogeneous to allow pure deterministic handling but exhibit too much organisation to enable pure statistical treatment.

A common way to define spatial organisation of a physical system is through its distance from the maximum entropy state (Kondepudi and Prigogine, 1998; Kleidon, 2012). Isolated systems, which do not exchange energy, mass, or entropy with their environment, evolve due to the second law of thermodynamics to a perfectly mixed "dead state" called thermodynamic equilibrium. In such cases, entropy is maximized and Gibbs free energy is minimized, because all gradients have been dissipated by irreversible processes. Hydrological systems are, however, open systems, as they exchange mass (water, chemicals, sediments, colloids), energy and entropy across their system boundaries with their environment. Hydrological systems may hence persist in a state far from thermodynamic equilibrium. They may even evolve to states of a lower entropy, and thus stronger spatial organisation, for instance through steepening of gradients, for example, in topography, or in the emergence of structured variability of system characteristics or network-like structures. Such a development is referred to as "self-organisation" (Haken, 1983) because local scale dissipative interactions, which are irreversible and produce entropy, lead to ordered states or dynamic behaviours at the (macro-) scale of the entire system. Self-organisation requires free energy transfer into the system to perform the necessary physical work, self-reinforcement through a positive feedback to assure "growth" of the organised structure/patterns in space, and the export of the entropy which is produced within the local interactions to the environment (Kleidon, 2012).

Manifestations of self-organisation in *catchment systems* are manifold. The most obvious one is the persistence of smooth topographic gradients (Reinhardt and Ellis, 2015; Kleidon et al. 2012), which reflect the interplay of tectonic uplift and the amount of work water and biota have performed to weather and erode solid materials, to form soils and create flow paths. Although these processes are dissipative and produce entropy, they nevertheless leave signatures of self-organisation in catchment systems. These are expressed, for instance, through the soil catena – a largely deterministic arrangement of soil types along the topographic gradient of hillslopes (Milne, 1931; Zehe et al., 2014) – and even more strongly through the formation of rill and river networks (Fig. 1) at the hillslope and catchment scales

(Howard, 1990; Paik and Kumar, 2010; Kleidon et al., 2013). These networks form because
flow in rills is, in comparison to sheet flow, associated with a larger hydraulic radius, which
implies less frictional energy dissipation per unit volume of flow. This causes higher flow
rates, which in turn may erode more sediment. As a result, these networks commonly increase
the efficiency in transporting water, chemicals, sediments and energy through hydrological
systems, which results also in increased kinetic energy transport through the network and
across system boundaries.
In contrast, the term self-organisation is rarely applied to *groundwater systems*, except in
the context of positive/negative feedbacks during processes of precipitation and dissolution
(e.g., Worthington and Ford, 2009). We argue, though, that the subsurface, too, displays some
characteristics of (partial) self-organisation. This is manifested, in particular, through
ubiquitous, spatially correlated, anisotropic patterns of aquifer structural and hydraulic
properties, particularly in non-Gaussian systems (Bardossy, 2006), as these have much
smaller entropy compared to spatially uncorrelated patterns. The emergence and persistence
of preferential pathways even in homogeneous sand packs (e.g., Hoffman et al., 1996; Oswald
et al., 1997; Levy and Berkowitz, 2003) is a striking example of formation of a self-organised
pattern of "smooth fluid pressure gradients".

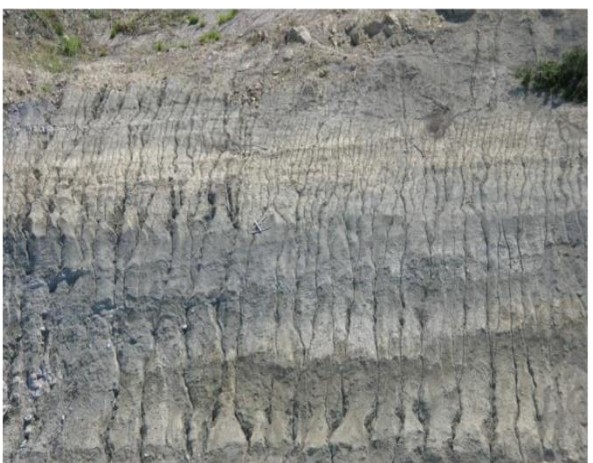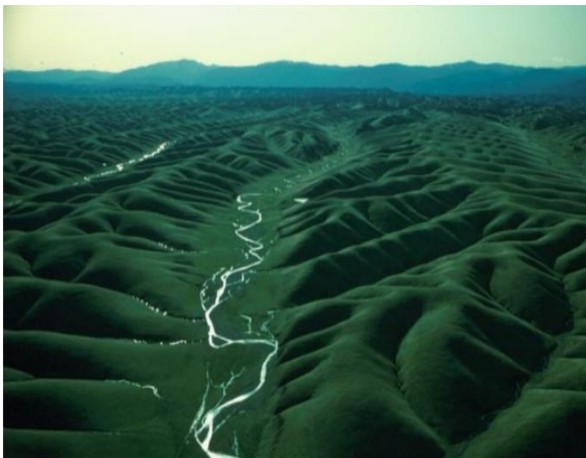

**Figure 1** Hillslope scale rill networks developed during an overland flow event at the
Dornbirner Ach Austria (left panel, we gratefully acknowledge the Copyright holder © Ulrike
Scherer KIT) and the South Fork of Walker Creek in California (right panel, we gratefully
acknowledge the Copyright holder © James Kirchner ETH Zürich)
Our general recognition is that hydrological systems exhibit – below and above ground –
both (structural, hydraulic and chemical) heterogeneity and signatures of (self-)organisation.
We propose that all kinds of preferential flow paths/flow networks veining the land surface
and the subsurface are prime examples of spatial organisation (Bejan et al., 2008; Rodriguez-
Iturbe and Rinaldo, 2001) because they exhibit, independently of their genesis, similar
topological characteristics. Our starting point to unify both water worlds is the recognition
that any form of preferential flow is a manifestation of self-organisation, because it hinders
perfect mixing within a system and implies a more "energy efficient" and hence faster
throughput of water and matter (Rodriguez-Iturbe et al., 1999; Zehe et al., 2010; Kleidon et
al., 2013). This general notion can be elaborated further by detailing the role of preferential
flow for transport of mass and chemical species, and related fingerprints in travel distances or
travel times, as well as for energy conversion and energy dissipation associated with flows of
water.

In terms models, hydrological modelling (and hydrological theory) attempts to predict
how processes described by equations evolve in and interact with a structured heterogeneous
domain (i.e., hydrological landscape). However, our key argument that both systems are
subject to similar manifestations of self-organisation does not imply proposed use of a single
model. Rather, we argue that similar conceptualisations and methods of quantification –
whether related to preferential flow paths, dynamics and patterning of chemical transport and
reactivity, or characterization in terms of energy dissipation and entropy production, for
example – can and should be applied to both catchment and groundwater systems, to the
benefit of both research communities. The main focus of this contribution is on the role of
heterogeneity and preferential flow and transport of water and chemical species. At a general
level, we show that preferential flow causes deviations from the maximum entropy state,
though these deviations have different manifestations depending on whether we observe
solute transport in space or in time. Based on this insight, we propose, essentially, that related
conceptualisations and quantitative characterisations can be unified in terms of a theory that is
applicable in catchment and groundwater systems, and thus connects these two "water
worlds".

We first discuss key features of fluid flow and chemical transport dynamics in these two
systems – catchments (including surface water) and groundwater – using the (often distinct)
terminology of each of these "water world" research communities. We outline the particular
questions, methods, limitations, and uncertainties in each "world" (Section 2). We then focus
on chemical transport, merging treatment of many of these dynamics in a proposed
quantitative framework, and providing specific examples (Section 3). More specifically,
Section 3 first defines specific conceptual and quantitative tools, and within this context,
introduces a continuous time random walk (CTRW) modelling framework with a clear
connection to microscale physics and to the well-known advection-dispersion equation.
Section 3 then offers new insights, in terms of contrasting power law and inverse gamma
distributions – used in the groundwater literature to describe different travel time distributions
that control long tailing in breakthrough curves – to gamma distributions used more often in
the surface water (catchment system) literature. This analysis is a basis for suggesting how
surface water systems (catchment response to chemical transport) can be treated within the
CTRW framework. Final conclusions and perspectives appear in Section 4. Throughout, we
attempt to offer an innovative synthesis of concepts and methods from the generally disparate
surface water (catchment hydrology) and groundwater research communities. Each
community has developed sophisticated modelling and measurement capabilities – which
have led to significant scientific advances over the last two decades – that could benefit the
other community and help address outstanding, unsolved problems.

Before proceeding, we emphasise that our use of the term "two water worlds" throughout
this paper is intended to highlight the disparate catchment and groundwater communities, and
is not used in the specific context of mobile-immobile water in the root zone (McDonnell,
2014), as discussed at the end of Section 3.1.

## 2  TWO WATER WORLDS – UNIQUE, DIFFERENT AND SIMILAR

### 2.1  Governing laws of fluid flow, the momentum balance and energy dissipation

In both water worlds, a major focus is on travel distances, as well as travel times (residence times) of water, as they provide the main link between water quantity and quality (Hrachowitz et al., 2016). Catchment hydrology deals also with extremes, i.e., floods and droughts, as well as land surface-atmosphere feedbacks, fluvial geomorphology and eco-hydrology.

From the outset, we recognise that predictions of water dynamics in catchment and aquifer systems require joint treatment of their mass, momentum and energy balances. Catchment science and modelling has, traditionally, a strong focus on catchment mass and (in part) energy balances, as evaporation and transpiration release energy in the form of latent heat to the atmosphere. The momentum balance is treated in an implicit conceptualised manner, as detailed below. Predictions of fluid flow in groundwater systems rely on the joint treatment of the mass and the stationary momentum balances using Darcy's law, while the energy balance appears at first sight of low importance.

Chemical transport and travel times through hydrological systems are, however, strongly related to both the momentum and the energy balances, because they jointly control the spectrum of fluid velocities and the direction of streamlines. The governing equations that characterise water flow velocities along the land surface and in groundwater systems are simplifications of the Navier-Stokes equations (Eq. 1), which describe the momentum balance of the fluid as an interplay of driving forces and hindering frictional forces:

$$\rho \frac{\partial \boldsymbol{v}}{\partial t} + (\boldsymbol{v} \cdot \nabla)\boldsymbol{v} = -\nabla \mathrm{p} - \rho \boldsymbol{g} + \eta \Delta^2 \boldsymbol{v} \qquad \text{(Eq. 1)}$$

where $\boldsymbol{v}$ (m s$^{-1}$) is the fluid velocity vector, $\boldsymbol{g}$ (m s$^{-2}$) the gravitation acceleration vector, $\rho$ (kg m$^{-3}$) the water density, and $\eta$ the dynamic viscosity (kg m$^{-1}$ s$^{-1}$) of the fluid.

### 2.1.1      Surface water flow and Manning's law

Overland and channel flow are driven by surface topography, or more precisely, by gravitational potential energy differences, but only minute amounts of these energy differences are converted into kinetic energy of the flow (Loritz et al., 2019), while the rest is dissipated. Surface water flow velocity is often characterised by Manning's law (Eq. 2), a steady-state, one-dimensional approximation of the Navier-Stokes equation that neglects inertial acceleration for the case of turbulent shear stress and thus turbulent energy dissipation. Fluid velocity grows proportional to the square root of the driving hydraulic head gradient; the latter corresponds to the potential energy of a unit mass of water:

$$\boldsymbol{v_{surface}} = -\frac{R^{\frac{2}{3}}}{n}\sqrt{2g\nabla_{x,y}(h+z)} = -\frac{R^{\frac{2}{3}}}{n}\sqrt{2g\nabla_{x,y}\Phi_H} \qquad \text{(Eq. 2)}$$

where $\boldsymbol{v_{surface}}$ (m s$^{-1}$) is the overland flow velocity vector, $R$ (m) the hydraulic radius defined as the ratio of the wetted cross-section $A_{\mathrm{wet}}$ (m$^2$) to the wetted perimeter $U_{\mathrm{wet}}$ (m), $n$ is Manning's roughness (m$^{-1/3}$), $z$ (m) is topographical elevation, $h$ (m) is depth of the flow, and $\Phi$ (m) is the total hydraulic head.

Moreover, as friction occurs mainly at the contact line between the fluid and the solid, the hydraulic radius R (m) can be used to scale the ratio between driving gravity force and the

hindering frictional dissipative force. Kleidon et al. (2013) classified this as a "weak form" of
dissipative interaction between fluid and solid. In this context, they showed that overland flow
in rills implies, due to the larger hydraulic radius, a smaller dissipative loss per unit volume
and thus a higher energy efficiency compared to sheet flow. Along the same line, they showed
that flow in a smaller number of wider channels is more efficient than flow in a higher
number of narrower channels. Both effects, flow in rills and channelling, lead to a higher fluid
velocity, and thus a higher power (kinetic energy) flux through the network. Note that a 10%
faster fluid velocity implies 30% more power as the latter grows with the cube of the fluid
velocity.

### 2.1.2   Subsurface flow and Darcy's law

Flow through subsurface porous media, on the other hand, is driven by the gradient in
total hydraulic head, reflecting differences in gravitational potential, matric potential and
pressure potential energies as described in the respective forms of Darcy's law (Eq. 3). The
latter is also a steady state, one dimensional approximation of the Navier-Stokes equation
neglecting the inertial terms. However, in this case flow is essentially laminar and dissipative
frictional losses in the porous medium are so much larger than in open surface flow, that
kinetic energy can be neglected. When solving the Darcy law (Eq. 3, first line) for the
interstitial travel velocities and defining the flow resistance as inverse hydraulic conductivity,
one obtains a form of the Darcy law (Eq. 3, second line) which is similar to Manning's law
(Eq. 2). The main difference arises from the different dependencies on the hydraulic head
gradient, reflecting the turbulent and laminar flow regimes, respectively:

$$\boldsymbol{q_{vadose}} = -k(\theta)\nabla(\psi + z) \,, \quad \boldsymbol{q_{gw}} = -k_s\nabla(H + z)$$

$$\boldsymbol{v_{vadose}} = -\frac{1}{\theta R(\theta)}\nabla\Phi_{vadose}, \quad \boldsymbol{v_{gw}} = -\frac{1}{\theta_s R_s}\nabla\Phi_{gw} \text{ (Eq. 3)}$$

$$R(\theta) = \left.1\middle/k(\theta)\right., R = \left.1\middle/k_s\right., \Phi_{vadose} = (\psi + z), \Phi_{gw} = (H + z)$$

where $\boldsymbol{q_{vadose}}$ and $\boldsymbol{q_{gw}}$ (m s$^{-1}$) are water flux vectors (filter velocities) in the partially saturated
and saturated zones, respectively, $\boldsymbol{v_{vadose}}$ and $\boldsymbol{v_{gw}}$ (m s$^{-1}$) are the respective interstitial travel
velocities, $\theta$ and $\theta_s$ are the soil water content (-) and the porosity (-), $k(\theta)$ and $k_s$ (m s$^{-1}$) are the
partially saturated and saturated hydraulic conductivity, $\psi$ (m) and $H$ (m) denote the capillary
pressure and pressure potentials, and $\Phi_{vadose}$ and $\Phi_{gw}$ are total hydraulic heads in the partially
saturated and saturated zones.
The strikingly high dissipative nature of porous media flow becomes obvious when
recalling that the driving matric potential gradients in the vadose zone are often orders of
magnitude larger than 1 m m$^{-1}$. This implies a capillary acceleration term much larger than
Earth's gravitational acceleration g (m s$^{-2}$), yet fluid velocities in the porous matrix are several
orders of magnitude smaller than in surface water systems. However, the generally much
slower fluid velocity in groundwater systems does not impose a slow hydraulic response time
during rainstorms; on the contrary, aquifers may release − almost instantaneously − "older",
pre-event water into a catchment outlet stream. This apparent paradox − referred to often as
the "old-new water paradox" (Kirchner, 2003) − is explained by propagation of pressure
waves. Shear or compression waves (or waves in general) transport momentum and energy

through continua *without an associated transport of mass or particles* (Everett, 2013; Goldstein, 2013); and group velocity (or "celerity") is many orders of magnitude larger than the fluid velocity in aquifer systems (McDonnell and Beven, 2014). Today, it is known that depending on landscape setting, antecedent wetness conditions, and the dominant runoff mechanisms, pre-event water fractions in storm runoff can vary from near zero to more than 60% of storm water having an isotopic signature different from that of rainfall (Sklash and Farvolden, 1979; Sklash et al., 1996; Blume et al., 2008).

### 2.1.3 Preferred flow paths as maximum power structures and non-Fickian transport

Flow velocity within subsurface preferential pathways (macropores, pipes, fractures) is known to be much faster than matrix flow (Beven and Germann, 1982, 2013). This is caused not only by the vanishing capillary forces, but also, largely, by the strong reduction in frictional dissipation in macropores compared to flow in the porous matrix. Viscous dissipation in preferential pathways occurs, similar to open channel flow, mainly at the contact line between fluid and solid, i.e., the wetted perimeter of the macropore, which implies – similar to the case of rill and river networks – a larger hydraulic radius and thus a much more energy efficient flow (Zehe et al., 2010). Darcy's law is hence inappropriate to characterise preferential flow (Germann, 2018). Clearly, rapid localised flow and transport in preferential pathways hinders the transition from imperfectly mixed stochastic advective transport in the near field to well mixed advective dispersive transport in the far field. Predictions of solute plumes and travel times in the near field are thus challenging as this requires detailed knowledge of the velocity field, while transport at the well-mixed Fickian limit depends on the average fluid velocity and the dispersion coefficient (Simmons, 1982; Sposito et al., 1986; Bodin, 2015).

Although the revisited laws, interactions and phenomena are well known, we suggest that the energy point of view yields a unifying perspective to explain why macropore, rill and river networks are the preferred (preferential) pathways for water flow on land and below. One might hence expect that water flows along the path of maximum power (Howard, 1990; Kleidon, 2013), which is the product of the flow velocity times the driving potential difference. The paths of maximum power correspond in the case of constant friction to the path of steepest descent in hydraulic head, while in the case of a constant gradient, it corresponds to the path of minimum flow resistance (Zehe et al., 2010). From the discussion above, we further conclude that catchment hydrology and groundwater hydrology are inseparable. We can neither separate a river from its catchment and its subsurface, nor an aquifer from the land surface and the catchment. Both stream flow response to rainfall and groundwater are composed of "waters of different ages", reflecting the ranges of overland flow, subsurface storm flow and base-flow contributions with their specific velocities, usually non-Fickian travel time distributions, and chemical signatures.

In the following, we elaborate briefly on the specific model paradigms in catchment and groundwater hydrology with an emphasis on preferential pathways for fluid flow and chemical transport, and on the resulting ubiquitous, anomalous early and late time arrivals of chemicals to measurement outlets.

## 2.2 Catchment hydrology from the water balance to solute transport

### 2.2.1 The catchment concept and the duality in water balance modelling

Catchment hydrology developed largely as an engineering discipline around traditional tasks of designing and operating reservoirs, flood risk assessment and water resources management (Sivapalan, 2018). Although the catchment concept is elementary to these tasks, we think it worthwhile to reflect briefly on it here. The watershed boundary delimits a control volume where the streamlines are expected to converge into the river network, hence ideally, the entire set of surface and subsurface runoff components feeds the stream. We can thus characterise the water balance of an ideally closed catchment control volume based on observations of rainfall input and stream flow response (with uncertainty). Even more importantly, the catchment water balance can be solved without an explicit treatment of the momentum balance, because flow lines end up in the stream.

This is a twofold blessing. First, hydrological models can be benchmarked against integral water balance observations. We posit that this unique property of catchments is *the* reason why integral conceptual hydrological models, which largely ignore the momentum balance, allow successful predictions of stream flow the catchment outlet (Sivapalan, 2018). As conceptual models directly address processes at the system level without accounting for subscale mechanistic reasons, their application is often referred to as "top down" modelling. The other end of the model spectrum consists of physics-based, spatially distributed models, originally proposed by the blueprint of Freeze and Harlan (1969), which follow a "bottom up" mechanistic paradigm. These models are thus also referred to as reductionist models. While the pros and cons of top down conceptual models and bottom up physics-based models have been discussed extensively, we agree with Hrachowitz and Clark (2017) that they offer complementary merits as detailed below. As an aside, it is interesting to reflect why conceptual models due not exist in the field of, e.g., meteorology. We suggest that this is because atmospheric flows are not governed by mechanisms similar to catchments, which implies that the amount of air mass flowing from one location to another cannot be predicted without knowing the flow lines.

### 2.2.2 Top down modelling of the catchment water balance

Top down conceptual hydrological models simulated water storage, redistribution and release within the catchment system through combination of non-linear and linear reservoirs, characterised by effective state variable and effective parameters and effective fluxes (Savenije and Hrachowitz, 2017). Due to their mathematical simplicity, conceptual models are straightforward to code. With the advent of combinatorial optimization methods for automated parameter search, and fast computers (Duan et al., 1992; Bardossy and Singh, 2008; Vrugt and Ter Braak, 2011), these models became, at first sight, also straightforward to apply. Automated, random parameter search led, however, to the discovery of the well-known equifinality problem – namely, that several model structures or parameter sets may reproduce the target data in an acceptable manner (Beven and Binley, 1992), within the calibration and validation period, but these models and parameter sets yield uncertain future predictions (e.g., Wagener et al., 2006). Equifinality and related parameter uncertainty arises from the ill-posed nature of inverse parameter estimation and from parameter interactions in the equations. While the first problem can tackled using multi-objective and multiresponse calibration (e.g.,

Mertens et al., 2004; Ebel and Logue, 2006; Fenicia et al., 2007), the latter is inherent to the model equations regardless of whether they are conceptual (as shown by Bardossy (2007) for the Nash cascade), or physically based (as shown by, e.g., Klaus and Zehe (2010) and Zehe et al. (2014)).

A well-known shortcoming of conceptual models is that their key parameters cannot be measured directly. This motivated numerous parameter regionalization efforts (He et al., 2011a) to relate conceptual parameters to measurable catchment characteristics, typically broadly available data on soils (including texture), land use, and topography. As a consequence, such functions have been derived successfully, for example, to relate parameters of the soil moisture accounting scheme to soil type and land use (as shown by, e.g., Hundecha and Bardossy, 2004; Samaniego and Bardossy, 2006; He et al., 2011b; Singh et al., 2016) or parameters of the soil moisture accounting of the mHm (Samaniego et al., 2010) to soil textural data. As such, relations are landscape-specific and they require a new calibration when moving to new target areas. This is of course possible if high quality discharge data are available. Yet, due to the incompatibility between the corresponding measurement and observations scales, these regionalisation functions are not straightforwardly explained using physical reasoning. This is true even if soil moisture accounting from soil physics is used, e.g., the Brooks and Corey (1964) soil water retention curve, as in the case of the mHm model.

A number of early efforts to meaningfully define hydrological response units for regional modelling of hydrological landscapes were reported by, e.g. Knudsen et al. (1986), Flügel (1995), and Winter (2001). Savenije (2010) and Fencia et al. (2011) significantly improved the link between conceptual models and landscape structure in their flexible model framework. The key idea is to subdivide the landscape into different functional units (plateaus, hillslopes, wetlands, rivers), and to represent each of them by a specific combination of conceptual model components to mimic their dominant runoff generation processes. Landscapes with different dominant runoff generation mechanisms are represented through an appropriate combination of these conceptual "building blocks" (Fenicia et al., 2014; Gao et al., 2014; Wrede et al., 2015) using suitable topographical signatures such as "Height Above Next Drainage" (Gharari et al., 2011) to estimate their areal share. This is a clear advantage that facilitates model calibration and reduction of predictive uncertainty.

The strength of integral conceptual models is their ability to provide parsimonious and reliable predictions of streamflow $Q$ ($m^3$ $s^{-1}$) directly at the catchment outlet. However, it is nevertheless not straightforward to apply the models for predictions of transport of tracers, and more generally chemical species through the catchment into a stream, as elaborated in the following.

### 2.2.3 Integral approaches to solute transport modelling in catchment hydrology

Predictions of solute transport require information about the spectrum of fluid velocities and travel distances across the various flow paths into the stream (we can usually neglect the travel time within the river network due to the much higher fluid velocities, as argued in Sect. 3.1). Such information can generally be inferred from breakthrough curves of tracers that enter and leave the system through well-defined boundaries, as shown for instance by the early work of Simmons (1982) and Jury et al. (1986), using transfer functions to model solute

transport through soil columns. The transfer function approach is based on the theory of linear
systems. This implies that the outflow concentration (volumetric flux-averaged concentration)
$C_{out}$ (kg m$^{-3}$) at time $t$ is, in case of steady-state water flow, the convolution of the solute
input time series $C_{in}$ with the system function $G$ (Green's function):

$$C_{out}(t) = \int_0^\infty G(t - \tau)\, C_{out}(\tau)d\tau \qquad \text{(Eq. 4)}$$

The transfer function is the system response to a delta function input. Note that Eq. 4
should in general be formulated for the input and output mass flows, which correspond to the
input/output concentration multiplied by the input/output volumetric water flows. It is
important to note in this context is that the average travel time through the system can be
calculated from the water flow and length of flow path, as the average travel velocity
corresponds to the flow divided by the wetted cross section of the soil column (see Eq. 3). The
latter implies that travel time distributions through partially saturated soils are transient and
hence constrained by the input time (Jury et al., 1986; Sposito et al. 1986). The well-known
fact that the flow velocity field changes continuously with changing soil water content
explains why transfer function approaches have been largely put aside in soil physics and
solute transport modelling in the partially saturated zone.
In the case of catchments, simulated runoff from conceptual hydrological models cannot,
unfortunately, be used to constrain the average transport velocity. This is simply because
conceptual models provide, by definition, no information about the wetted cross of the flow
path through the catchment, and the latter determines essentially the average fluid velocity $v$
from simulated total runoff $Q$. The fact that the simple equation $Q = v_{transport} A_{wet}$ has an
infinite solution space, if $A_{wet}$ is unknown, is also a major source of equifinality. This was
shown by Klaus and Zehe (2010) and Wienhöfer and Zehe (2014), using a physically-based
hydrological model to investigate the role of vertical lateral preferential flow paths of
hillslope rainfall runoff response. These authors found that several network configurations
matched the observed flow response equally well: some configurations consisted of a small
number of larger macropores of higher conductance, while others consisted of a higher
number of less conductive macropores. Overall, these configurations yielded the same
volumetric water flow, but they performed rather differently with respect to simulation of
solute transport. An even larger challenge for transport modelling through catchments arises
from the fact that the distribution of flow path lengths is even more difficult to constrain,
compared to a soil column.
Despite these challenges, the tracer hydrology community made considerable progress in
understanding catchment transit time distributions and predicting isotope or tracer
concentrations in streamflow (Harman, 2015). Initially, stable isotopologues of the water
molecule and other tracers gained attention as they allow a separation of the storm hydrograph
into pre-event and event water fractions using stable end member mixing (Bonell et al., 1990;
Sklash et al., 1996). Today isotopes of the water molecules and water chemistry data are used
as a continuous source of information to infer travel time distributions of water through
catchments (McGlynn et al., 2002; McGlynn and Seibert, 2003; Weiler et al., 2003; Klaus et
al., 2013). Early attempts to predict tracer concentrations in the stream relied on the same kind
of transfer functions as outlined in Eq. (4) for soil columns. Hence, they naturally faced the
same problems of state and thus time-dependent travel time distributions (Hrachowitz et al.,
2013; Klaus et al., 2015; Rodriguez et al., 2018). More recent approaches rely on age ranked
storage as a "state" variable in combination with StoreAgeSelection (SAS) functions for
stream flow and evapotranspiration to infer their respective travel time distributions (Harmann
et al. 2015; Rinaldo et al., 2015). Aged ranked storage needs to be inferred from solving the
Master equation, i.e., the catchment water balance for each time and each age. This can be
done either by using conceptually modelled or observed discharge and evapo-transpiration,
and it requires a proper selection of the functional form of the SAS functions and optionally
their time-dependent weights (Rodriguez and Klaus, 2019). Related studies rely on a single or
several gamma distributions (Hrachowitz et al., 2010; Klaus et al., 2015; Rodriguez and
Klaus, 2019), others used the beta distribution (van der Velde et al., 2012) or piece-wise linear
distributions (Hrachowitz et al., 2013, 2015).
Here we propose that the continuous time random walk (CTRW) framework from the
groundwater "world" has much to offer to catchment travel time modelling (as detailed in
Sect. 3). We show that, in particular, the inverse gamma distribution may offer a useful
alternative that offers the asset of a clear connection to microscale physics and the well-
known advection-dispersion equation, which is used in bottom up modelling (Sect. 2.2.4). In
this context, it is interesting to recall that catchments were modelled as time-invariant linear
systems for a considerable time, since the unit hydrograph was introduced by Sherman
(1932). While the effect of precipitation was calculated using runoff coefficients, the
streamflow response was simulated by convoluting effective precipitation with the system
function, i.e., the unit hydrograph. The "Nash" cascade of linear reservoirs was a popular
means to describe the unit hydrograph in a parametric form, and it is well known that the
latter is mathematically equivalent to a gamma distribution (Nash, 1957). As streamflow
response of the catchment is affected largely by surface and subsurface preferential pathways,
which cause non-Fickian transport, one might hence wonder whether a gamma distribution
function is an ideal choice to represent the fingerprint of preferential flow.

### 2.2.4    Bottom up modelling of the catchment water balance

The blueprint of a physically based hydrological, introduced by Freeze and Harlan (1969),
has found manifold implementations. Physically-based models like MikeShe (Refsgaard and
Storm, 1995) or CATHY (Camporese et al., 2010) typically rely on the Darcy-Richards
concept for soil water dynamics (Eq. 3), the Penman–Monteith equation for soil-vegetation-
atmosphere exchange processes, and the Manning's equation for estimating overland and
stream flow velocities (Eq. 2).
Each of these approaches is naturally subject to limitations, reflecting our yet imperfect
understanding, and suffers from the limited transferability of their related parameters from
idealised, homogeneous laboratory conditions to heterogeneous and spatially organised
natural systems (Grayson et al., 1992; Gupta et al., 2012). In this context, the Darcy-Richards
model receives by far the strongest criticism (Beven and Germann, 2013), simply because the
underlying assumption regarding the dominance of capillarity-controlled diffusive flow, under
local equilibrium conditions, is largely inappropriate when accounting for preferential flow.
The Darcy model is hence incomplete when accounting for infiltration (Germann, 2018) and
preferential flow, and several approaches have been proposed to close this gap. These range

from the early idea of (a) stochastic convection assuming no mixing at all (Simmons, 1982), to (b) dual-permeability conceptualizations relying on overlapping, exchanging continua (Šimunek et al., 2003), to (c) spatially explicit representations of macropores as connected flow paths (Vogel et al., 2006; Sander and Gerke, 2009; Zehe et al., 2010; Wienhoefer and Zehe, 2014; Loritz et al., 2017), and to (d) pore-network models based on mathematical morphology (Vogel and Roth, 2001). An alternative approach to dealing with preferential flow and transport employs Lagrangian models such as SAMP (Ewen, 1996a,b), MIPs (Davies and Beven, 2012; Davies et al., 2013), and LAST (Zehe and Jackisch, 2016; Jackisch and Zehe, 2018; Sternagel et al., 2019).

Reductionist models are, despite the challenge to represent preferential flow and transport, indispensable tools for scientific learning. They particularly allow exploration of how distributed patterns and their spatial organisation jointly controls distributed state dynamics and integral behaviour of hydrological systems (Zehe and Blöschl, 2004). Related studies range from, e.g., the investigation of (a) how changes in agricultural practices affect the stream flow generation in a catchment (Pérez et al., 2011), (b) the role of bedrock topography for runoff generation using (Hopp and McDonnell; 2009) at the Panola hillslope and the Colpach catchment (Loritz et al. 2017), or (c) the role of vertical and lateral preferential flow networks on subsurface water flow and solute transport at the hillslope scale (Bishop et al., 2015; Wienhöfer and Zehe; 2014; Klaus and Zehe; 2011; Klaus and Zehe, 2010), including the issue of equifinality. Setting up a physically-based model, however, requires an enormous amount of highly resolved spatial data, particularly on subsurface characteristics. Such data sets are rare, and the "hunger" for data in such models risks a much higher structural model uncertainty. On the other hand, these models offer also greater opportunities for constraining their structure using multiple data orthogonal to discharge (Ebel and Logue, 2006; Wienhöfer and Zehe, 2014).

Another asset of reductionist models is their thermodynamic consistency, which implies that energy conversions related to flow and storage dynamics of water in the catchment systems are straightforward to calculate (Zehe et al, 2014). This offers the opportunity to test the feasibility of thermodynamic optimality as constraint for parameter inference (Zehe et al.; 2013), the latter is rather challenging when using conceptual (Westhoff et al., 2013, Westhoff et al., 2016). More recent applications demonstrated, in line with this asset, new ways to simply distributed models without lumping, which allowed the successful simulation of the water balance of a 19 km$^2$ large catchment using a single effective hillslope model (Loritz et al., 2017). The key to this was to respect energy conservation during the aggregation procedure, specifically through derivation of an effective topography that conserved the average distribution of potential energy along the average flow path length to the stream; and through a macroscale effective soil water retention curve that conserved the relation between the average soil water content and matric potential energy using a set point scale retention experiments (Jackisch 2015; Zehe et al., 2019).

Along similar lines, Loritz et al. (2018) showed that simulations using a fully distributed setup of the same Colpach catchment using 105 different hillslopes yielded strongly redundant contributions of stream flow (Fig. 2). The used the Shannon entropy (Shannon, 1948, defined in Eq. 6 in section 2.4) to quantify the diversity in simulated runoff of the hillslope ensemble at each time step. They found that although the entropy of the ensemble

was rather dynamic in time, it never reached the maximum value. Note that an entropy maximum implies that hillslopes contribute in a unique fashion, while a value of zero implies that all hillslope yield a similar runoff response. They further showed that the fully distributed model, consisting of 105 hillslopes, can be compressed to a model using 6 hillslopes with distinctly different runoff responses, without a loss in simulation performance. Based on these findings, they concluded that spatial organisation leads to emergence of functional similarity at the hillslope scale, as proposed by Zehe et al. (2014). This in turn explains why conceptual models can be reasonably applied as most of the spatial heterogeneity in the catchment seems to be irrelevant for runoff production. However, this is not the case, when it comes to transport of chemicals as elaborated in the next section.

In accord with Hrachowitz and Clark (2017), we conclude that top down and bottom up models indeed have complementary merits. Moreover, we propose that the applicability of conceptual models at larger scales arises from the fact that spatial organisation leads in conjunction with the strongly dissipative nature of hydrological process to the emergence of simplicity at larger scales (Savenije and Hrachowitz, 2017; Loritz et al., 2018).

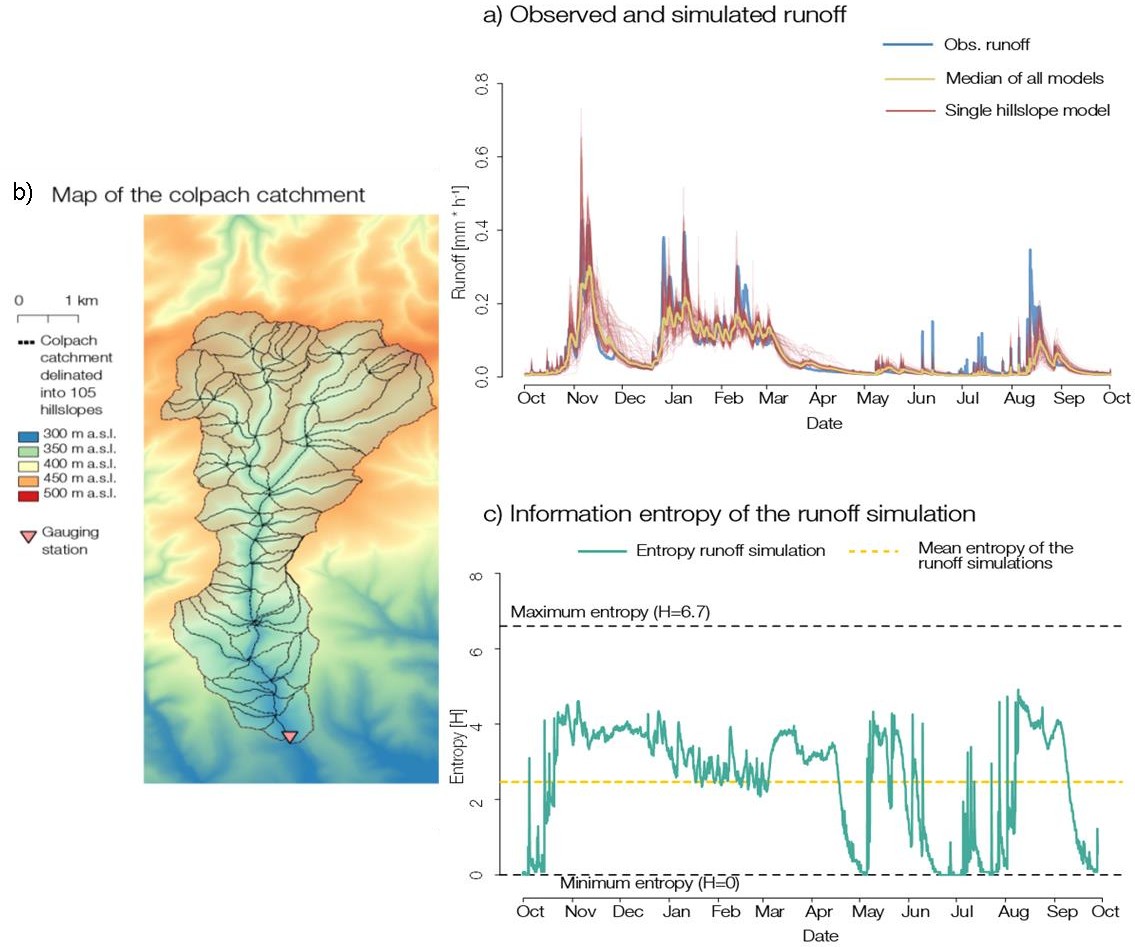

**Figure 2 (a)** Observed and simulated runoff of the Colpach catchment. The red lines correspond to individual hillslope models and the yellow line to area weighted median of all hillslopes. **(b)** Map of the Colpach catchment and the 105 different hillslopes. **(c)** Shannon entropy in turquoise for the runoff simulations as well as the corresponding mean. © Ralf Loritz KIT, from Loritz et al. (2018).

## 2.3   Distributed solute transport modelling – the key role of the critical zone

Reductionist physically models are straightforwardly to couple with the advection-dispersion equation (compare Eq. 11 in Sect. 3) or particle tracking schemes to simulate transport of tracers and reactive compounds through the critical zone into groundwater or along the surface and through the subsurface into the stream.

The soil-vegetation-atmosphere-transfer system (SVAT-system), or in more recent terms, the "critical" zone, is the mediator between the atmosphere and the two water worlds. This tiny compartment controls the splitting of rainfall into overland flow and infiltration, and the interplay among soil water storage, root water uptake and groundwater recharge. Soil water and soil air contents control $CO_2$ emissions of forest soils, denitrification and related trace gas emissions into the atmosphere (Koehler et al. 2010; Koehler et al, 2012), as well as biogeochemical transformations of chemical species.

Partly saturated soils may, depending on initial their state and structure, respond with preferential flow and transport of contaminants and nutrients through the biological most active topsoil buffer (Flury et al., 1994, 1995; Flury, 1996; McGrath et al., 2008, 2010; Klaus et al., 2014). Rapid transport operates within strongly localized preferential pathways such as root channels, cracks, worm burrows or within connected inter-aggregate pore networks which "bypass" of the soil matrix continuum (e.g., Beven and Germann, 1982; Blume et al., 2009; Wienhöfer et al., 2009; Beven and Germann, 2013). The well-known fingerprint of preferential flow is a "fingered" flow pattern, which is often visualised through dye staining or two-dimensional concentration patterns in vertical soil profiles (Fig. 3). These reveal imperfectly mixed conditions in the near field, which implies that the spatial concentration pattern deviates from the well mixed Fickian limit over a relatively long time. The latter corresponds in the case of a delta input to a Gaussian distribution of travel distances at a fixed time, where the centre of mass travels with the average transport velocity while the spreading of the concentration grows linearly with time proportional to the macrodispersion coefficient (Simmons, 1982; Bodin, 2015). Note that according to Trefry et al. (2003), this Gaussian travel distance corresponds to a state of maximum entropy. Preferential flow hence implies a deviation from this well mixed maximum entropy state, which cannot be predicted with the advection-dispersion equation (e.g., Roth and Hammel, 1996). A recent study (Sternagel et al., 2019) revealed that even double domain models such as Hydrus 1D may fail to match the flow fingers and/or long-time concentration tails in tracer profiles. Frequently, the partially saturated region of the subsurface is simply too thin to allow perfectly mixed Gaussian travel distances to be established; hence non-Fickian transport in the critical zone is today regarded as being the rule rather than the exception.

Germany, van Schaik et al. (2014)   Chile, Blume et al. (2009)  Austria, Wienhöfer et al. (2009)

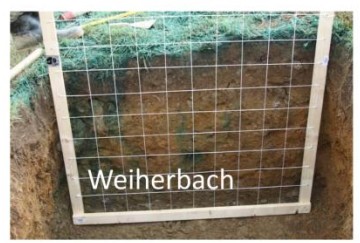 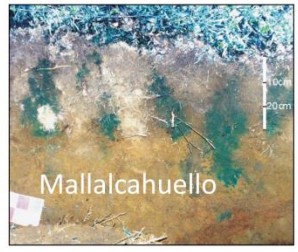 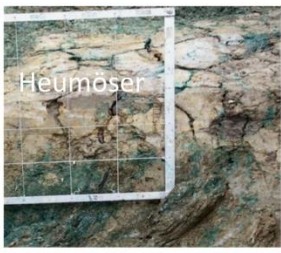

Germany, Zehe and Flühler (2001)      Switzerland, Flury et al. (1994)

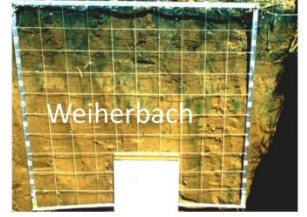 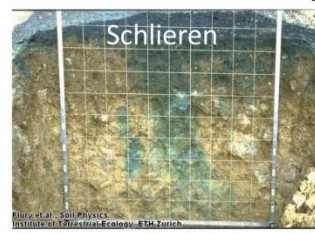 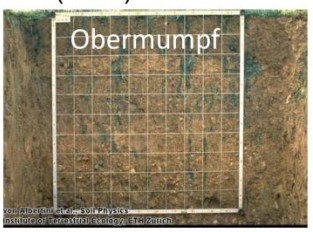

**Figure 3** Finger flow pattern revealed from standardized dye staining experiments for a transport time of 1 day; images where generously provided by Flury et al. (2004; 2005; Copyright © 1994, 1995 the American Geophysical Union) for Switzerland, Blume et al. (2009, Copyright © Theresa Blume) for Chile, Wienhöfer et al. (2009, Copyright © Jan Wienhöfer KIT) for Austria, and Zehe and Flühler (2001, Copyright © Erwin Zehe KIT) and van Schaik et al. (2014, Copyright © 2013 John Wiley & Sons, Ltd.) for the German Weiherbach.

Because preferential transport leads to strongly localized accumulation of water and chemical species, preferential pathways are potential biogeochemical hotspots. This is particularly the case for biopores such as worm burrows and root channels. Worm burrows provide a high amount of organic carbon and worms "catalyse" microbiological activity due to their enzymatic activity (Bundt et al., 2001; Binet et al., 2006; Bolduan and Zehe, 2006; van Schaik et al., 2014). Similarly, plant roots provide litter and exude carbon substrates to facilitate nutrient uptake. Intense runoff and preferential flow events optionally connect these isolated "hot spots" to lateral subsurface flow paths such as a tile drain network or a pipe network along the bedrock interface, and thereby establish "hydrological connectivity" (Tromp-van Meerveld and McDonnell, 2006; Lehmann et al., 2007; Faulkner, 2008). The onset of hydrological connectivity comprises again a "hot moment" as upslope areas and, potentially, the entire catchment start "feeding" the stream with water, nutrients and contaminants (Wilcke et al., 2001; Goller et al., 2006).

The critical zone, furthermore, crucially controls the Bowen ratio (the partitioning of net radiation energy into sensible and latent heat), and soil water available to plants is a key controlling factor. The residual soil water content is not available for plants, as it is generally stored in fine pores subject to very high capillary forces. Isotopic tracers have been fundamental to unravelling water flow paths in soils, using dual plots (Benettin et al., 2018; Sprenger et al., 2018), and to distinguish soil water that is recycled to the atmosphere and released as stream flow (Brooks et al., 2010; McDonnell, 2014).

Further to the above points, it is noted that laboratory and numerical studies of multiple
cycles of infiltration-drainage of water and chemicals into a porous medium demonstrate
clearly the establishment of stable "old" water clusters/pockets, and even a "memory effect"
(Kapetas et al., 2014), which remain even with multiple cycles of "new" water infiltration
(Gouet-Kaplan and Berkowitz, 2011). These pore-scale studies are in qualitative (and semi-
quantitative) agreement with studies at the *field scale,* which show similar retention behaviour
of bromide (introduced during the first infiltration cycle) after multiple infiltration-drainage
cycles (Turton et al., 1995; Collins et al., 2000). As a consequence, when each cycle of
infiltration contains water with a different chemical signature, stable pockets of water can be
established with highly varying chemical composition. We hence emphasise that mobile and
immobile waters sustaining evaporation and stream flow – and the chemical species they
contain – exist at a continuum of scales from the pore to the field level. Thus, rather than
attempting to delineate pockets of less and more mobile water at each scale – separating these
pockets at the pore, the column, the meter, the 10 meter, and the field and catchment scales –
we instead suggest recognising and delineating an "overall effect" of separation between
"old" (immobile) and "new" (mobile) waters at a given "effective" scale of interest, which
integrates over all such old and new waters. As we discuss in detail at the end of Sect. 3.1, and
thereafter, then, we argue that it is a more effective approach to consider chemical transport as
following *distributions of travel distances and residence times,* which can then characterized
by various (often power law) probability density functions.

## 2.4   Groundwater systems

As noted in Sect. 1, analysis of groundwater systems has developed largely independently
of investigation of catchment systems, although it, too, developed originally as a large
deterministic engineering discipline around the traditional task of water supply for domestic
and agricultural use. It was only in the 1980's that "stochastic" (probabilistic and statistical)
techniques began to be implemented extensively, to account for the many uncertainties
associated with aquifer structure and hydraulic properties that control the flow of
groundwater. In parallel, significant interest (and concern) with water quality and
environmental contamination in groundwater systems only entered the research community's
consciousness in the 1980s, although some pioneering laboratory experiments and field
measurements were initiated from the late 1950s.
It is worth noting, too, that the methods and models applied in groundwater research
developed independently and separately from research on catchment systems (Sect. 1). The
only partial connection or "integrator" has traditionally been with aquifer connections to the
vadose zone (or critical zone, discussed in Sect. 2.3). Another connection between surface
water and groundwater systems, though not generally recognized as such, has been analysis of
water flow, and to a lesser extent chemical species transport, in the hyporheic zone. The
hyporheic zone can be defined as the region of sediment and subsurface porous domain below
and adjacent to a streambed, which enables mixing of shallow groundwater and surface water.
(e.g., Haggerty et al, 2002).
To quantify chemical transport, landmark laboratory experiments (e.g., Aronofsky and
Heller, 1957; Scheidegger, 1959) measured breakthrough of conservative (non-reactive)
chemical tracers through columns of sand. These measurements underpinned theoretical

developments, based also on concepts of Fickian diffusion, which led to consideration of the classical advection-dispersion equation. Since that time, the advection-dispersion equation – and variants of it – have been used extensively to quantify chemical transport in porous media. However, as thoroughly discussed in Berkowitz et al. (2006), solutions of the advection-dispersion equation have repeatedly demonstrated an inability to properly match results of extensive series of laboratory experiments, field measurements, and numerical simulations. These findings naturally lead to the conclusion that the conceptual picture underlying the advection-dispersion equation framework is insufficient; as detailed in Sect. 2.2, the soil physics community arrived at a similar conclusion. Stochastic variants of the advection-dispersion equation, and implementation of multiple-continua, advection-dispersion equation formulations (including mobile-immobile models) have been used to provide insights into factors that affect chemical transport – particularly given uncertain knowledge of detailed structural and hydraulic aquifer properties – but they have been largely unable to capture measured behaviours of chemical transport. This observation is largely in line with what we reported for the critical zone.

The first key is to recognize that heterogeneities are present at all scales in groundwater systems, from sub-millimetre pore scales to the scale of an entire aquifer. Indeed, use of the term "heterogeneities" refers to varying distributions of structural properties (e.g., porosity, presence of fractures and other lithological features), hydraulic properties (e.g., hydraulic conductivity), and – in the case of chemical transport (a general term used here and throughout to denote migration of chemical (and/or microbial) components) – variations in the biogeochemical properties of the porous domain medium. The second key is to recognize that these variations in distributions, at all scales, deny the possibility of obtaining complete knowledge of the aquifer domain in which fluids and chemical species are transported. A third key, when considering chemical transport (and transport of stable water molecule isotopes), is to recognize that chemical species are subject to several critical transport mechanisms and controls, in addition to advection, that do not affect flow of water – molecular diffusion, dispersion, and reaction (sorption, complexation, transformation) – so that chemical migration through an aquifer is influenced strongly by aquifer heterogeneities and initial/boundary conditions. Extensive analysis of high-resolution experimental measurements and numerical simulations of transport demonstrate that small-scale heterogeneities can significantly affect large-scale behaviour, and that small-scale fluctuations in chemical concentrations do not simply average out and become insignificant at large scales.

As discussed in the preceding sections, preferential pathways are ubiquitous and affect both water and chemical species, resulting from system heterogeneity. To be more specific, (local) hydraulic conductivities vary in space over orders of magnitudes, even within distances of centimetres to meters, and these variations ultimately control patterns of fluid and chemical movement. The resulting patterns of movement in these systems involve highly ramified preferential pathways for water movement and chemical migration. To illustrate these points, consider the hydraulic conductivity ($K$) and preferential pathway maps shown in Fig. 4a; see Edery et al. (2014) for full details.

Figure 4a shows a numerically-generated, two-dimensional domain measuring $300 \times 120$ discretized into grid cells of uniform size (0.2 units). The $K$-field shown here was generated as a random realization of a statistically homogeneous, isotropic, Gaussian $\ln(K)$ field, with

ln($K$) variance of $\sigma^2 = 5$. Fluid flow through this domain was solved at the Darcy level by
assuming constant head boundary conditions on the left and right boundaries, and no-flow
horizontal boundaries; the hydraulic head values determined throughout the domain were then
converted to local velocities, and thus streamlines. Conservative chemical transport was
determined using a standard Lagrangian particle tracking method, with $10^5$ particles
representing the dissolved chemical species. Particles advanced by advection along the
streamlines and molecular diffusion (enabling movement between streamlines), to generate
breakthrough curves (concentration vs. time) at various distances along the domain. Figure 4b
shows particle pathways through the domain, wherein the number of particles visiting each
cell is represented by colours. The emergence of distinct, limited particle preferential
pathways from inlet boundary to outlet boundary is striking. Notably, too, there are significant
regions that remain free of particles (the white regions in Fig. 4b), and preferential pathways
are confined and converge between low conductivity areas. Even more striking is set of even
sparser preferential pathways shown in Fig. 4c: here, only cells, which were visited by at least
0.1% of all injected particles, are shown. In other words, 99.9% of all chemical species
migrating through the domain shown in Fig. 4a advance through a limited number and spatial
extent of preferential pathways. It is significant, too, that the preferential pathways comprise a
combination of higher conductivity cells the paths, but also some low conductivity cells, as
reported also in Bianchi et al. (2011); see Sect. 3.1 for further discussion of this behaviour.
(a)

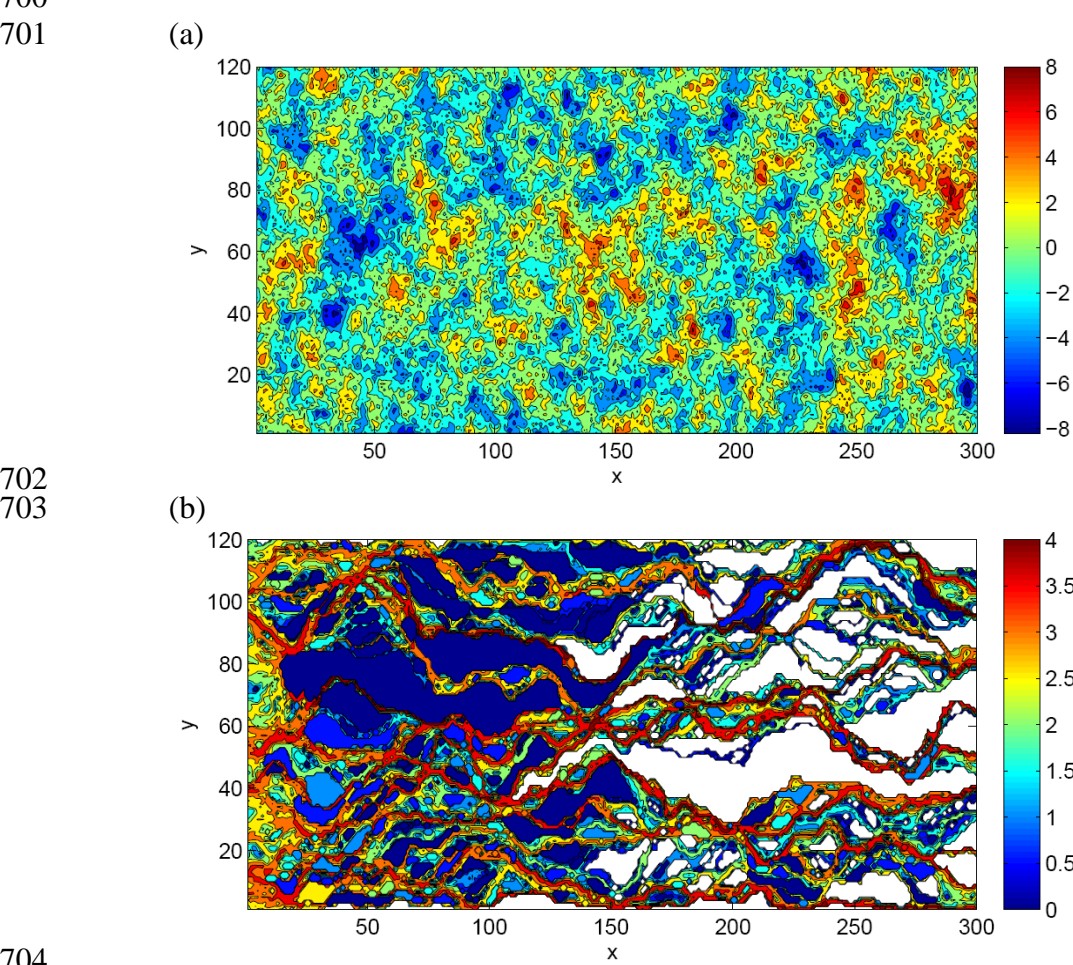

(b)


(c)

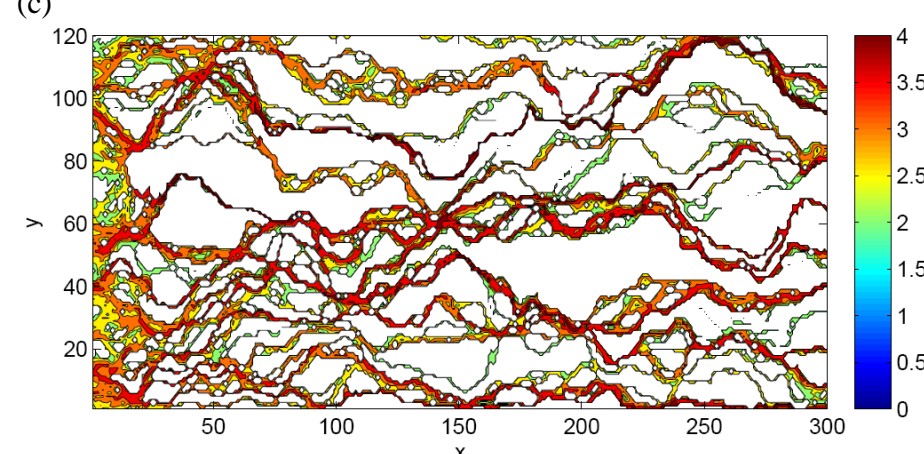

**Figure 4 (a)** Spatial map showing a sample hydraulic conductivity (*K*) field generated
statistically (right side bar shows scale of ln(*K*)). **(b)** Spatial map showing particle paths
through the domain, for overall hydraulic gradient (water flow) from left to right. "Particles"
representing dissolving chemical species are injected along the left vertical boundary and
followed through the domain. White regions indicate where *no* particles "visit" (interrogate)
the domain. Blue regions have only a small number of particle visitations. Red regions have
significant particle visitations. Note that the colour bar is in log10 number of particles. **(c)**
Spatial map showing particle paths *preferential* particle paths, defined as paths through cells
(underlying subdivisions in the domain, each with a different *K* value as shown in plot (a)
above) that each contain a "visitation" of a minimum of 0.1% of the total number of particles
in the domain. Note that the colour bar is in log10 number of particles (after Edery et al.,
2014; Copyright 2014, with permission from the American Geophysical Union).
Thus, it is clear that the groundwater systems incorporate regions of water – distributed
throughout the domain – that may have very different chemical signatures, even in close
proximity to each other. Moreover, these regions can be relatively stable over time, modified
only by the extent of chemical diffusion into and out of the "immobile" regions.
In accord with our definition of spatial organisation in Sect. 1, we propose the use of
Shannon entropy *H* (bits) to quantify the degree of spatial organisation in the flow pattern in
Fig. 4c. To this end, we define the discrete probability density distribution to find a particle in
a grid element, $\Delta y_i$, at the inlet (*x*=0) and at the outlet (*x*=300) of the flow domain, based on
the numbers of particles that entered/left the domain through the corresponding grid cells
divided the total number of particles that entered/left the domain $N_{in}/N_{out}$, as follows:

$$p(x = 0, \Delta y_i) = \frac{n(\Delta y_i, x=0)}{N_{in}}; \; p(x = 300, \Delta y_i) = \frac{n(\Delta y_i, x=300)}{N_{out}} \qquad \text{(Eq. 5)}$$

where $p(x=0, \Delta y_i)$/ $p(x=300, \Delta y_i)$ are probabilities that particle entered/left the domain at $\Delta y_i$,
$n(x=0, \Delta y_i)$/ $n(x=300, \Delta y_i)$ are the numbers of particles that entered/left the domain at $\Delta y_i$.
Using these probability distributions, we calculate the respective Shannon entropy values
defined as follows:
$$H = -\sum p_i log_2(p_i) \qquad \text{(Eq. 6)}$$

The Shannon entropy of the uniform input distribution, with 6.9 bits, corresponds to an
entropy maximum. Preferential flow reduced this to $H = 3.58$ bits at the outlet, which reflects
a release of chemicals that is much more organised in space. Note that a well-mixed
advective-dispersive pattern would maximise the entropy at the outlet, as the concentration
would be constant along the $y$ coordinate. Considering now arrival times of chemical species
at the domain outlet boundary, Fig. 5 shows the relative concentration ($C/C_o$) vs. time –
breakthrough curves – for three degrees of domain heterogeneity ($\ln(K)$ variance). (The well-
mixed case would maximise the entropy at the outlet, corresponding to a CTRW fit with $\beta = 2$
in Fig. 5.) It is evident that the chemical transport in this domain displays "non-Fickian" (or
"anomalous") transport, in the sense that late-time (long tail) arrivals are registered at the
measurement plane. Furthermore, Fickian-based advection-dispersion equation models clearly
fail to quantify such behaviour (Fig. 5). However, Fig. 5 shows solutions – based on the
continuous time random walk (CTRW) framework – that do effectively describe the chemical
transport. The CTRW framework and governing transport equations are detailed in Sect. 3.3.

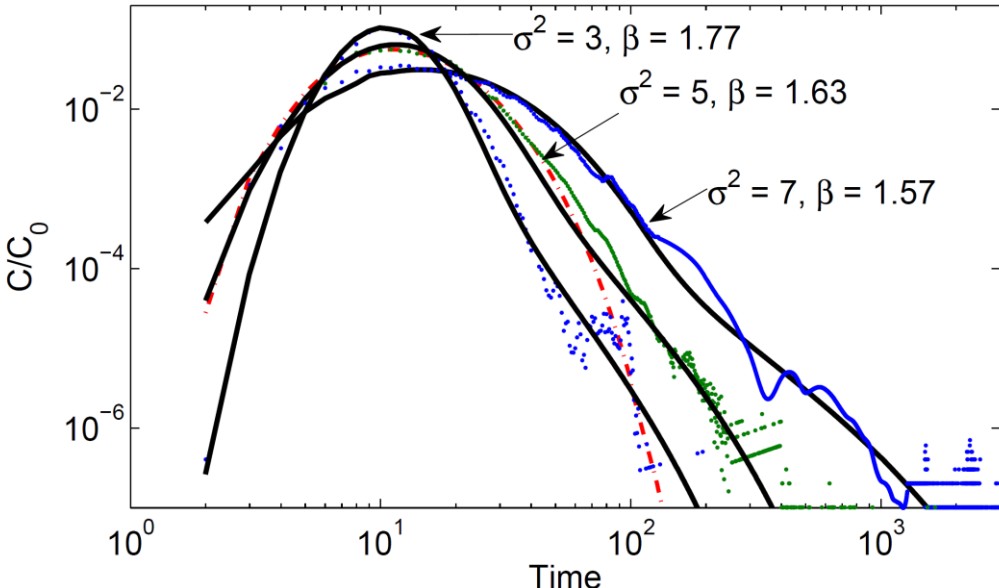

**Figure 5** Breakthrough curves (points) for three $\ln(K)$ variances ($\sigma^2 = 3,5,7$; 100 realizations
each), at the domain outlet ($x = 300$ length units), and corresponding CTRW fits (curves).
Also shown is a fit of the advection-dispersion equation (dashed-dotted curve), for $\sigma^2 = 5$.
See Sect. 3.3 for further discussion and explanation of $\beta$. All values are in consistent, arbitrary
length and time units (after Edery et al., 2014; Copyright 2014, with permission from the
American Geophysical Union).

## 3 MERGING TREATMENT OF SURFACE WATER AND GROUNDWATER SYSTEM TRANSPORT DYNAMICS

### 3.1 Conceptual pictures, travel times, and mixtures of water with different chemical signatures

Clearly, any quantitative model of fluid flow and chemical transport in a catchment must
first define a conceptual picture. In the context of the discussion in Sects. 2 and 3 that led us
to this point, we require a picture that accounts naturally for overland and interacting
subsurface flow and transport, recognizing the ubiquity of preferential pathways and a broad
(and often different) distributions of fluid and chemical travel times. Moreover, any such
conceptual picture also requires definition of the available measurement benchmark against
which a quantitative model can be compared. In the case of catchments, a common
measurement is that of chemical arrival times at a downstream sampling point in a catchment
stream that drains and exits the catchment. Thus, the dynamics of fluid flow and chemical
transport in a fully three-dimensional (or simplified two-dimensional overland) catchment are
often represented by measurements in an effective, spatially averaged one-dimensional
system. (Of course, higher resolution, multidimensional (in space) measurements, if available,
should also be considered in a quantitative model!)

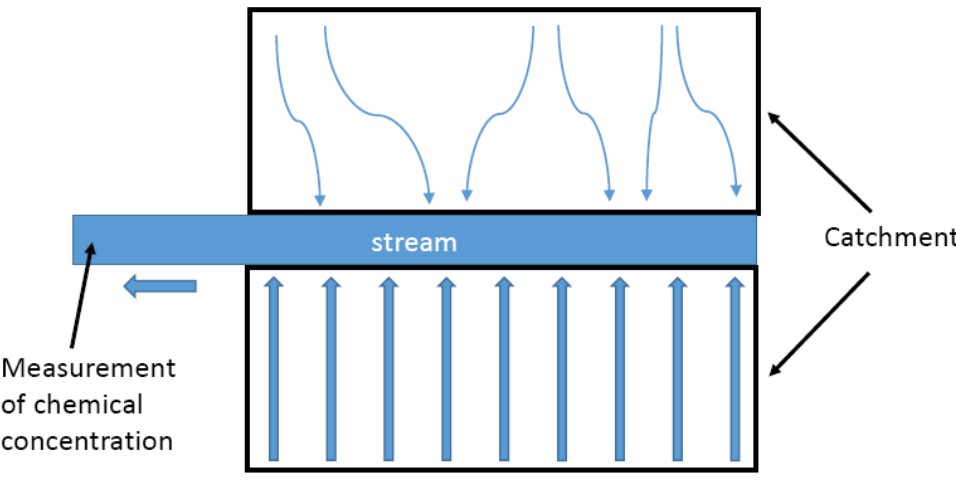

(a)

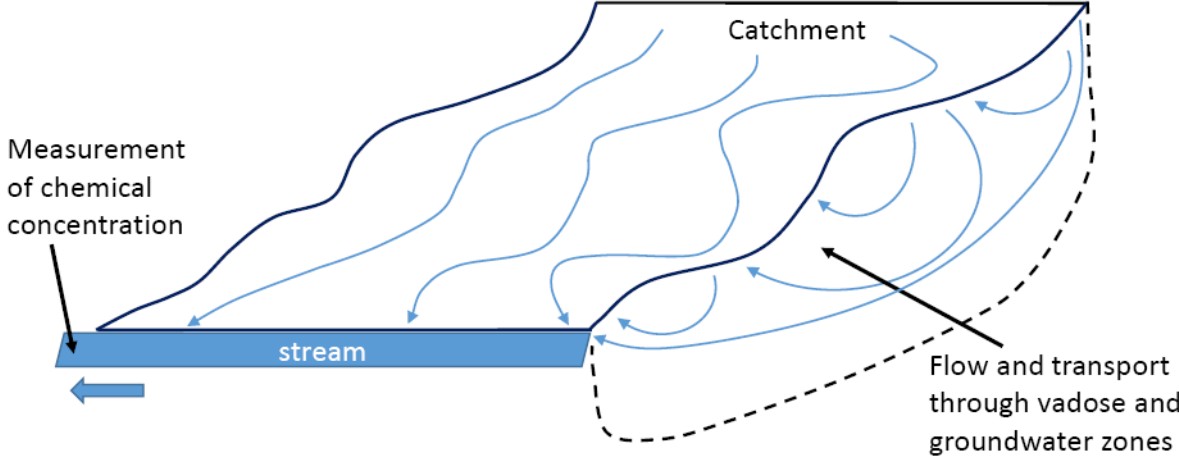

(b)
**Figure 6** Conceptual pictures of water flow and chemical transport in catchments under a
pulse of rainfall over the entire catchment. Each curved arrow (or idealized straight arrow)
indicates a different path, each of which embodies different travel times through the system
until reaching the stream. Note that each preferential pathway carrying water and chemical
species may be purely overland, or include interactions and advance within soil layers
(partially saturated, or vadose, zone) and saturated groundwater systems. **(a)** Schematic
showing idealized 2D catchment area. Arrows through two rectangular regions of catchment
indicate a range of preferential pathways carrying water and chemical species. **(b)** Schematic
showing idealized 3D catchment area, under a pulse of rainfall over the entire catchment.

Figures 6a and 6b show, schematically, 2D and 3D conceptualizations of preferential
pathways, with associated varying travel times through the catchment, for both fluid flow and
chemical transport. We stress here – and as discussed below in Sect. 3.3 – that the larger
scale, effective (or "characteristic", or average) fluid velocities and chemical species transport
velocities need not be identical. For example, using a conceptual mixing model, Hrachowitz
et al. (2015) showed that chloride transport can be slower than water transport. In fact, these
two velocities are rarely the same, as a consequence of the ubiquity of preferential pathways
for water and migrating chemical species in any surface water and/or soil-aquifer domain.
Because of these pathways, regions of higher and lower hydraulic conductivity (fluid and
chemical mobility) – and thus the entire system – interrogated by water and chemical species
differ. While both water molecules and chemical species are subject to diffusive and
dispersive transport mechanisms, in addition to advection, these effects are clearly identifiable
for chemical species, while they are undistinguishable for individual water molecules. Thus
the effects of diffusion and dispersion on "bulk water" transport, e.g., into and out of low
conductivity zones, are invisible and irrelevant, while chemical species retained in these same
zones can have a major impact on the overall (and "average", centre of mass) advance of a
chemical plume. These effects are also visible and relevant for isotopes of the water molecule,
as deuterium and tritium are subject to self-diffusion in water. The latter implies that isotope
concentrations between old and new water pockets in the subsurface might mix diffusively,
even when there is no physical mixing between these waters. Hence, the relation between
water age and its isotopic decomposition is not straightforward.
The conceptual picture discussed here is our basis for arguing that we should expect to
find distributions of travel times and mixtures of water with different chemical signatures, *at*
*all scales*. Moreover, these considerations align well with our reflections in Sect. 2 and key
studies in catchment hydrology, which clearly recognize the occurrence of wide distributions
of water and chemical travel times, and long-term chemical persistence in water catchment
storage (e.g., Niemi, 1977; Botter et al., 2010, 2011; Hrachowitz et al., 2010; McDonnell and
Beven, 2014; Kirchner, 2016).
As pointed out in Sect. 2, several studies in recent years have specifically reported the
presence of water bodies (or pockets, or regions, depending on scale), with different chemical
compositions and isotopic signatures, that are in close proximity or even "overlapping" (in
some sense). Some authors use the term "two water worlds" – immobile and mobile – in this
context (e.g., McDonnell, 2014) to describe the different sources of water returned to the
atmosphere by vegetation transpiration and released to streams; we stress again that our use of
the term in this paper highlights the different catchment hydrology and groundwater
communities and associated research tools. In light of the discussion in Sect. 2, we stress here
that the conceptual picture to explain spatially and temporally varying chemical compositions
(in subsurface, soil, sediment and aquifer systems), and associated uptake by vegetation, is
subtle. We question the conceptualization of two (or more) *separate, fully compartmentalized*
mobile and immobile regions of water and chemicals. We argue that mobile and immobile
regions are more appropriately considered as overlapping continua or ensemble/effective
averages, as those are found at all scales from pores to hundreds of meters (e.g., Turton et al.,
1995; Collins et al., 2000; Gouet-Kaplan and Berkowitz, 2011; recall Sect. 2.3). With the
occurrence of mixtures of travel times and waters having different chemical signatures at all
scales, we argue that it is preferable to think in terms of time, such that there is a range of
overlapping temporal (transition time) distributions that each contribute to the overall, large-
scale fluid flow and chemical transport. This leads naturally to the CTRW framework.
## 3.2  Space vs. time: the travel time perspective of transport
It is critical to point out that in the figures shown above in Sects. 2.3 and 2.4, the
*residence times* of water and chemicals are the key factors that determine transport behaviour.
This leads to the continuous time random walk (CTRW) framework, which operates more (or
at least equally!) in terms of time than in terms of space (see Sect. 3.2). To introduce CTRW,
in the context of the pathway "self-organisation" shown in Fig. 5c, we demonstrate the
importance of thinking in terms of *time* rather than *space*. Consider the simple example of
driving a distance of 100 km; we consider a scenario in which we travel 50 km at 1 km/h, and
then 50 km at 99 km/h. The average speed of travel, in terms of *space* (distance), is
determined as follows: given that we travelled 50 km at each of two speeds, the average speed
is $(1 + 99) / 2 = 50$ km/h. Thus, with this calculation, the total time to travel 100 km "should"
be 2 h. However, the *actual* time taken to travel this distance − 50 km at 1 km/h, and then 50
km at 99 km/h − is 50.5 h. In other words, traditional (but incorrect!) conceptual spatial
thinking highlights the erroneous effects of focusing only on *spatial* heterogeneity and
quantification based only on spatial characteristics.
In a similar analogy, it is sometimes faster to pass through a bottleneck region (e.g., drive
for a short time through a very narrow and slow road) to ultimately reach a fast highway,
rather than to travel at medium speed along a road for an entire journey.
Another aspect related to misplaced emphasis on spatial heterogeneities, is also noted
here. Referring again to the preferential pathways show in Fig. 4c, it is seen that these
pathways actually contain some low hydraulic conductivity ($K$) regions as well! This can be
explained most easily, conceptually, in terms of one-dimensional pathways. Consider a
number of high and low $K$ cells in series, [3 3 3 3 3] vs. [6 6 1 6 6], where the
effective/average $K$ is given by the harmonic mean. While a [3 3 3 3 3] series may appear to
enable a greater volumetric flow rate than a [6 6 1 6 6] series, due to the "bottleneck" low $K$
value in the centre, both series in fact have the same harmonic mean (=3) and conduct fluid
equally well.
A similar argument can be applied to analysis of land topography and surface water flow.
The "high resistance" (in principle, but not necessarily), localized small 'humps of roughness
elements', and surface tension effects − analogous to the low $K$ cells given in the previous
paragraph − can be overcome, to allow development of preferential pathways that do not
always follow the path of steepest descent in terms of surface topography. There are thus
small bypassing effects. Moreover, there is flow/transport from land surface into the
subsurface (e.g., hyporheic zone), which also "bypasses" localized small "humps" in the land
surface and allows fluid connection/communication further downstream (along a pathway).
As a consequence, we argue that it is misleading to place undue focus on the high resistance
(or surface "hump") bottlenecks; rather, it should be recognized that entire "high K" or
"potential" regions for flow are often unsampled or barely sampled by flowing water and
chemicals, at least over moderate time scales.
To further expand on the link between spatial and temporal heterogeneity, we point out
that the key is to think in space-time and complementary manifestations of heterogeneity of
preferential flow. We already showed that a heterogeneous preferential flow pattern implies
that chemical species leave the system at distinct locations, which implies a strong reduction
in Shannon entropy, as shown in Sect. 2.4 for the example of Edery et al. (2014). When
observed at a fixed outlet, these heterogeneous flow patterns translate into signatures of the
breakthrough curve. Again, this can be quantified through the corresponding deviations from
a Fickian breakthrough curve, which is the maximum entropy travel time distribution,
reflecting well-mixed, advective-dispersive transport (Tefry et al., 2003). The overall key
messages of Sect. 3 are that (a) CTRW is consistent with the advection-dispersion equation
and advances beyond it, particularly in terms of capturing dispersion and tailing effects, and
(b) the power law exponent is related to porous media characteristics as well as the flow
conditions, although this relation is not unique. Nevertheless, the opportunity arises to at least
partly constrain spatial signatures of the subsurface from temporal ones with uncertainty. This
non-uniqueness is another manifestation of the inherent equifinality problem, when reviewing
model concepts in catchment science in Sect. 2.1.
In the next section, we adopt a temporal framework to introduce continuous time random
(CTRW) theory, which is the basis of our proposed means to unify quantification of
groundwater and surface water transport dynamics.

## 3.3  Continuous Time Random Walks: Theory

Preferential flow leads to non-Fickian (or "anomalous") travel time distributions,
characterised by rapid breakthrough and/or long tailing of chemical species through
heterogeneous domains. The CTRW framework is well suited to deal with this in a manner
that is consistent with microscale physics, and it steps beyond the advection-dispersion
equation approach. This might also offer opportunities to understand SAS from a bottom up
perspective, as age ranked storage relates to the integral of the travel time distribution across
all ages.
Detailed descriptions of CTRW can be found in, e.g., Berkowitz et al. (2006, 2016). Here,
we present only a brief outline of the essential elements. The CTRW framework is based on
direct incorporation of the distribution of flow field fluctuations and thus of the fluctuations in
concentrations of transported chemicals. As such, the CTRW is a time-nonlocal approach that
can quantify chemical transport over a range of length (and time) scales, and address other
processes such as chemical reactions.
From a microscale of view, "particles", representing dissolved chemical species, are used
to treat chemical transport; each particle undergoes spatiotemporal transitions – "transitions
(or steps) in a random walk" – that encompass both displacement due to structural
heterogeneity and the time taken to make each particle movement. Unlike other approaches,
the formulation focuses on retaining the full distribution of transition times. Thus, CTRW
defines a probability density function (PDF), $\psi(\mathbf{s}, t)$, of a random walk that couples the spatial
displacement **s** and time *t* of the transition. As shown in Dentz et al. (2008), it is convenient
and generally applicable (but not obligatory) to use the decoupled form $\psi(\mathbf{s}, t) = p(\mathbf{s})\psi(t)$,
where $\psi(t)$ is the probability rate for a transition time *t* between sites, and $p(\mathbf{s})$ is the
probability distribution of the length of the transitions. We stress here that the particle
*transition* time distribution represents the PDF of times for any given particle transition over
the distance **s**, while the *travel* time distribution – also called a "first passage time
distribution" – discussed above and below is the PDF of arrival times (an "overall response")
through a catchment, soil column, or aquifer at a measurement point or plane. A breakthrough
curve, representing the concentration of all particles arriving at a control/measurement point
(or plane) over time, can then be determined by calculating the average travel (first passage)
times of all particles exiting boundary of the flow domain. Thus, the *transition* time
distribution – however chosen – is the PDF underlying the resulting solution (which can be
characterized in terms of the breakthrough curve, as well as *travel* time, or first passage time,
distribution, as well as in terms of spatial profiles and moments) of the governing transport
equation; see Sect. 3.4 for further discussion. [Note: Regarding first passage time distributions
and breakthrough curves, a subtlety must be kept in mind, namely, that the breakthrough
curve is equal to the first passage time distribution if one measures it at an absorbing
boundary ("exiting the flow domain" could be represented by an absorbing boundary).
Otherwise, the flux-averaged concentration is obtained from the net flux across a boundary
(Simmons, 1982; or Appendix of Dentz et al., 2004). Nevertheless, the analytical expressions
for the first passage time distribution and flux concentration are equal under certain boundary
conditions.]
The defining transport equation is equivalent to a generalized master equation (GME),
which is essentially a mass balance equation in space and time. Using a Taylor expansion, the
GME can be transformed into the continuum version (ensemble-averaged system) of the
CTRW, in the form of an integro-partial differential equation:

$\qquad \frac{\partial c(\mathbf{s},t)}{\partial t} = \int_0^t dt' M(t-t')[-\mathbf{v}_\psi \cdot \nabla \tilde{c}(\mathbf{s},t') + \mathbf{D}_\psi : \nabla\nabla \tilde{c}(\mathbf{s},t')]$  (Eq. 7)

for the normalized concentration c(**s**, *t*), where *M* is a memory function, the transport velocity
$\mathbf{v}_\psi$ and the generalized dispersion $\mathbf{D}_\psi$ are defined in terms of the first and second moments of
$p(\mathbf{s})$, and with the dyadic symbol : denoting a tensor product. In Laplace space, (1) becomes

$\qquad u\tilde{c}(\mathbf{s},u) - c_o(\mathbf{s}) = -\tilde{M}(u)[\mathbf{v}_\psi \cdot \nabla \tilde{c}(\mathbf{s},u) - \mathbf{D}_\psi : \nabla \tilde{c}(\mathbf{s},u)]$  (Eq. 8)

where the memory function $\tilde{M}(u) \equiv \bar{t}u\tilde{\psi}(u) / [1 - \tilde{\psi}(u)]$, $\bar{t}$ is a characteristic time, and with
~ denoting Laplace space and *u* denoting the Laplace variable. Note that this continuum
formulation contains a nonlocal-in-time convolution, in terms of the memory function.
In contrast to the classical advection-dispersion equation (see Eq. (11), below), the
"transport velocity," $\mathbf{v}_\psi$ is in principle distinct from the "average fluid velocity," **v**. This is
because chemical transport is "clearly identifiable", subject to diffusive and dispersive
mechanisms (recall the discussion following Fig. 6), so that the effective, overall transport
(i.e., a "characteristic" velocity) of chemical may be faster or slower than the average fluid

velocity. We point out, moreover, that residence times are a key characterisation, as they generally differ for water and chemical species. To illustrate, it is sufficient to recognise that the preferential flow paths themselves are generally stable when the overall hydraulic gradient changes (unless dealing with significant changes or turbulent flow), so that the residence time dictates the relative influence of diffusion and chemical movement into and out of less mobile zones, which ultimately affects breakthrough curves (Berkowitz and Scher, 2009).

It is critical to recognize that the occurrence of "rare events" – even a small proportion of chemical species migrating extremely slowly in some regions, and/or being repeatedly trapped and released of slow regions over a series of spatial transitions – are sufficient to lead to anomalous transport and extremely long "average" chemical transport times (Berkowitz et al., 2016). Thus, it is important to differentiate between "average" (recall Sect. 3.1) and "effective" transport of "most" particles. Indeed, we emphasise, too, that the effects of these "rare events" are deeply significant: they do not simply average out, but rather propagate to larger time and space scales.

With the decoupled form $\psi(\mathbf{s}, t) = p(\mathbf{s})\psi(t)$, the transition time distribution, $\psi(t)$, is thus at heart of the CTRW framework, and its form determines the memory function; the role of $p(\mathbf{s})$ on non-Fickian transport is relatively insignificant as long it has a compact (finite) range (Dentz et al., 2008). As discussed in detail (e.g., Berkowitz et al., 2006, 2016), it is expedient to define (t) as a truncated power law (TPL), which enables an evolution to Fickian behaviour:

$$\psi(t) = \frac{n}{t_1} \exp(-t/t_2) / (1 + t/t_1)^{1+\beta} \qquad (\text{Eq. 9})$$

for $0 < \beta < 2$, with the normalization constant

$$n \equiv (t_1/t_2)^{-\beta} \exp(-t_1/t_2) / \Gamma(-\beta, t_1/t_2) \qquad (\text{Eq. 10})$$

and with $\Gamma(-\beta, t_1/t_2)$ denoting the incomplete Gamma function (Abramowitz and Stegun, 1970). This functional form of $\psi(t)$ has been particularly successful in interpreting a wide range of laboratory and field observations, as well as numerical simulations. We chose the characteristic time appearing in the memory function to be $t_1$, which represents the onset of the power law region The truncated power law form of $\psi(t)$ behaves as a power law proportional to $(t/t_1)^{-1-\beta}$ for transition times in the range $t_1 < t < t_2$; $\psi(t)$ decreases exponentially for transition times $t > t_2$. Thus, the TPL enables quantification of non-Fickian transport, with a finite (sufficiently small) $t_2$, it facilitates (where appropriate) a longer-time, smooth evolution to Fickian transport. We note, too, that the CTRW framework also simplifies (e.g., Berkowitz et al., 2006, 2016) to specialized subsets of non-Fickian transport behaviour embodied within, e.g., multirate mass transfer (Haggerty and Gorelick, 1995) and fractional derivative (Zhang et al., 2009) formulations.

It is important to recognize, too, that specification of a pure exponential form for $\psi(t)$, namely $\psi(t) = \lambda \exp(-\lambda t)$, with mean $1/\lambda$, and/or choice of $\beta > 2$, reduces the CTRW transport Eq. (7) to the classical advection-dispersion equation, given in a general form as

$$\frac{\partial c(\mathbf{s},t)}{\partial t} = -\mathbf{v}(\mathbf{s}) \cdot \nabla c(\mathbf{s},t) + \nabla \cdot [\mathbf{D}(\mathbf{s})\nabla c(\mathbf{s},t)]$$ (Eq. 11)

where $\mathbf{v}(\mathbf{s})$ is the velocity field and $\mathbf{D}(\mathbf{s})$ is the dispersion tensor.

It is thus clear that the power law exponent $\beta$ in $\psi(t)$ characterises the local disorder of the system and the degree of non-Fickian transport as an integral, temporal fingerprint in the breakthrough curves. This reflects the effect of a strongly localised preferential movement of chemical species on travel times (recall Fig. 4), caused by the pattern of local driving gradients and hydraulic conductivity. Because the particle movement is clearly organised in *space*, we suggest that this might be seen as self-organisation: local disorder is manifested in deviation from advective-dispersive transport, which leads to *non-local*, organised dynamic behaviour in *time* at the system scale. This implies that the CTRW framework provides a means to quantify the integral, temporal fingerprint of spatially organised preferential flow through the power law exponent $\beta$ and the related distance from a Gaussian travel time distribution.

The CTRW transport equation, in partial differential equation form, can be solved in Laplace space (Cortis and Berkowitz, 2005) as well as in real space (Ben-Zvi et al., 2019). One can also solve the transport equation by implementing various particle-tracking formulations. This was done, for example, to obtain the fits to the long-tailed breakthrough curve displayed in Fig. 6. Particle tracking (PT) approaches offer an efficient numerical tool to treat a variety of chemical transport scenarios (for both conservative and reactive chemical species). They are particularly well-suited to accounting for pore-scale to column-scale dynamics. "Particles" (representing chemical mass) advance by sampling transitions in space and time from the associated CTRW distributions. We emphasize that this PT approach can be employed to treat both advection-dispersion equation (Fickian, normal transport) and CTRW (non-Fickian, anomalous transport) formulations, via appropriate choice of (exponential or power law, respectively) $\psi(t)$.

The efficacy and relevance of the CTRW framework has been demonstrated extensively for subsurface chemical transport (Berkowitz et al., 2006, 2016; Berkowitz and Scher, 2009; and references therein), from pore to aquifer scales, on the basis of extensive numerical simulations, laboratory experiments and field measurements. The formulation for chemical transport is general and robust over length scales ranging from pore to field, for different flow rates within the same domain, for chemically-reactive species, and even for time-dependent velocity fields (Nissan et al., 2017).

To conclude this section, and bridge to discussion that follows in the next section, we point out here that, the *curved power law* form can in some cases be a useful representation rather than the truncated power law (TPL), Eq. (9), as shown by Nissan and Berkowitz (2019). In this case, we write $\psi(t)$ as a curved power law function (Chabrier, 2003)

$$\psi(t) = C_1\, t^{-1-\beta} \exp(-t^*/t)$$ (Eq. 12)

where $C_1 \equiv (t^*)^\beta/\Gamma(\beta)$, is the normalization constant of the probability density function and $\Gamma$ is the Gamma function. Here, $t^*$ (a characteristic time) controls the exponential increase, while $\beta$ accounts for the power law region. It is important to note that this curved power law is an

*inverse gamma distribution*, with shape parameter $\beta$ and scale (or rate) parameter $t^*$. Note that
unlike the TPL in Eq. (9), notwithstanding the exponential term in Eq. (12), there is no cut-off
time that enables a transition to Fickian transport. These perspectives will be discussed in
detail in Sect. 3.4.

### 3.4   Continuous Time Random Walks: Application to surface water systems

In the context of our discussion in Sects. 2 and 3.1, recognizing that dynamics of chemical
transport in surface water and groundwater systems are at least phenomenologically and
functionally/dynamically similar over enormous spatial and temporal scales, we argue there
that simulations and analysis using the CTRW framework are meaningful and applicable also
to quantifying the (anomalous) dynamics of chemical transport in surface water systems. In
both surface water and groundwater systems, there is always "unresolved heterogeneity" (e.g.,
hydraulic conductivity, structure) at all scales. Fluid and chemical inputs range from being
reasonably well-defined to unknown (e.g., in terms of location and extent of a subsurface
contamination leak, areal extent and space-time heterogeneities of rainfall and related stable
isotope concentrations), while outputs may also be reasonably well-defined to unknown (e.g.,
arrival times of a chemical species to a monitoring point downstream, such as a stream gauge,
near surface spring or tile drain outlet). As a consequence, efforts to delineate preferential
flow paths and quantify chemical transport must be "adjusted" (or "be appropriate") to the
level of knowledge and spatial/temporal resolution.

More specifically, we note that the preferential pathways shown in Fig. 4b,c are
(phenomenologically, at least) similar to those of surface water systems shown in Fig. 1,
while the (temporal) breakthrough curves in Fig. 5 are similar to those determined at stream
gauges and tile drain outlets. Clearly, in surface water systems, and throughout small,
intermediate and large scales, there are stable regions of "water pockets" (less mobile water)
that can be distinguished by strongly varying chemical (ionic, isotopic) compositions. The
presence of tributaries leading to rivers in catchments demonstrates clear channelling effects
and the establishment of preferential pathways (Sect. 2).

Before discussing chemical transport and considering CTRW applications in the context
of surface water systems, we emphasise – as described early in Sect. 3.3 – the
interrelationship between transition time distributions, travel time distributions, and
breakthrough curves. The *transition* time distribution, as used particularly in the context of
particle tracking and random walk model formulations, is the underlying ("building block")
characterization of chemical movement in the domain. In other words, the *transition* time
distribution controls the nature of the overall transport. The *travel* time distribution is
obtained as the normalized histogram of the travel times (which can be based on the *transition*
time distribution) over all flow paths, or in other words, the travel time is the sum of the
individual transition times and the distribution is obtained by sampling over all travel times.
[Note: If one integrates the travel time distribution over all particles entering the system (in
space and in time), for a step input, one obtains the cumulative breakthrough curve (*c* vs. *t*).
The relation between flux concentrations, pulse inputs, and breakthrough curves, relative to
the first passage time distribution for a homogeneous medium, is discussed in Section 3.1 of
Dentz and Berkowitz, 2003).]
In the context of these three types of quantification of chemical movement, and in light of
consideration of Eqs. (3) and (6) and the analysis to follow below, we stress the fundamental
importance of the underlying transition time distribution in quantifying chemical transport
through an aquifer or catchment. Common formulations of the governing transport equation,
particularly the advection-dispersion equation and many variants thereof, do not include an
explicit accounting of the transition or travel time distributions. However, as seen from the
discussion of Eq. (11), an underlying exponential transition time distribution in the CTRW
transport equation leads to the advection-dispersion equation with a Gaussian breakthrough
curve. In sharp contrast, in the case of a power law transition time distribution that scales as
scales as $t^{-1-\beta}$, such as given in Eqs. (9) and (12), the resulting breakthrough curve for a
point/pulse input also scales as $t^{-1-\beta}$, as a direct consequence of the generalized central limit
theorem (e.g., Dentz and Berkowitz, 2003, Eqs. (73) and (82)). For a step input, the scaling is
$t^{-\beta}$, because it can be obtained from the point by integration in time.
CTRW has also been applied in some partially saturated soil-water systems, which further
strengthens the connection of CTRW to surface water systems; as discussed in Sects. 2 and
3.1 (Figs. 4a,b), surface water flow and associated chemical transport are not purely overland
processes, but involve coupled interactions with the partially saturated (vadose) zone (Sect.
2.3) and groundwater zone (Sect. 2.4).
Indeed, CTRW methods (and subsets) have already been applied in some sense, at least
qualitatively, to interpret anomalous transport in various surface water system scenarios. For
example, Boano et al. (2007) used CTRW to quantify chemical transport in a stream,
accounting for fluid-chemical interactions with the underlying sediment (i.e., the hyporheic
zone). Other studies have recorded power law and related multirate rate mass transfer
dynamics for chemical transport in stream and catchment systems (e.g., Haggerty et al., 2002;
Gooseff et al., 2003). These authors note, in particular, that the hyporheic zone exhibits an
enormous range of time scales over which chemical exchange can occur, with significant
amounts of chemical species being retained over extremely long times.
However, while full application of CTRW to catchment-scale surface water systems has
not been reported to date, there are additional strong indications that it is applicable. We point
out two key aspects to support this claim, from the catchment hydrology literature. As
discussed in Sect. 2.2.3, previous studies used a gamma distribution to parameterize travel
time distributions (e.g., Hrachowitz et al., 2010), while more recent studies use a single or
several gamma distributions to characterise StorAgeSelection functions of stream flow and
evaporation. The gamma distribution, used particularly in connection with arrival times of
stable isotopes at a catchment outlet (= river outlet, measurement control plane) – i.e., as a
*travel* time distribution – has been applied to describe the superposition of different functions
to account for time dependence (e.g., Hrachowitz et al., 2010). Related directly to this point,
too, are unit hydrograph analyses that were used in the past to describe runoff concentration
and flood routing, through a Nash Cascade, which is essentially a gamma distribution, as
discussed also in Sect. 2.2.3. We now focus on this aspect in detail.
The gamma distribution is given by


$$P(t) = C_2\, t^{-1+\beta} \exp(-t\, t^*),$$                    (Eq. 13)


where $C_2 \equiv (t^*)^\beta / \Gamma(\beta)$, or, equivalently (and for comparison to Eq. (12)),

$$P(t) = C_3 \, t^{-1+\beta} \exp(-t/t^*), \qquad\qquad \text{(Eq. 14)}$$


where $C_3 \equiv 1/[(t^*)^\beta \, \Gamma(\beta)]$. The gamma distribution describes processes for which the waiting
times between Poisson distributed events are important.
In light of Sect. 2, and the discussion of transition and travel time distributions in Sect. 3.3
and above, we consider what underlying *transition* time distribution leads to a gamma
distributed *travel* time. Given that a sum of gamma distributed random variables can also be
gamma distributed, the choice of a gamma distribution for both *transition* and *travel* time
distributions is convenient. -et al., 2017).
Indeed, in terms of transition time distributions, let us compare the gamma distribution in
the form of Eq. (14) to the inverse gamma distribution as shown in Eq. (12). Aside from the
normalisation coefficients, the inverse gamma and gamma distributions shown in Eqs. (12)
and (14) differ in two fundamental ways – the power law (exponent of $t$) terms, $t^{-1-\beta}$ vs. $t^{-1+\beta}$
and the exponential terms, $\exp(-t^*/t)$ vs. $\exp(-t/t^*)$, respectively. We stress again, as explained
in Sect. 3.2, that the *inverse* gamma distribution is a power law distribution (without an
exponential cut-off time to allow transition to Fickian transport), and thus one form of
transition time distribution $\psi(t)$ in the CTRW formulation.
We plot in Fig. 7a the truncated power law, curved power law (inverse gamma) and
gamma (*transition* time) distributions, $P(t)$, for the specific parameters $\beta = 1.5$, $t_1 = 1$, $t_2 = 10^3$,
$t^* = t_1$. We plot in log-log scale to emphasize the long-time portion of the transition time
distribution. Figure 7b shows the same curves plotted on a linear scale, to contrast the fact that
linear plots (noting the short time scale on the $x$-axis) do not illustrate the long-time
contributions, which can have a critical effect on the overall transport behaviour. Clearly,
from Fig. 7b, the gamma distribution has does not include the possibility of long times; it has
an exponential cut-off to Gaussian behaviour at times larger than $t^*$, as the exponential term
dominates the power law term when $t \gg t^*$. However, note that the power law is $t^{-1+\beta}$ rather
than $t^{-1-\beta}$. The inverse gamma distribution, on the other hand, does not display an exponential
cut-off, but has the same $t^{-1-\beta}$ power law scaling as the TPL.
We thus conclude (recall also the conceptual picture and discussion in Sect. 3.1) that
although there is no universally "right" or "wrong" choice, the gamma (*transition* time)
distribution does not generally appear as a suitable "candidate" to quantify chemical transport
in surface water systems, notwithstanding its empirical use in the literature. We suggest that
the CTRW framework (Sect. 3.3) rests on a more physically justified conceptual picture and
corresponding, coherent and robust mathematical formulation; other such frameworks and
transition time distributions can of course also be considered, if justified physically. The
choice of a truncated power law or inverse gamma (*transition* time) distribution is largely a
function of scale. The inverse gamma distribution may better suit pore-scale (microscale)
domains, where the peak of the function is important, and where ergodicity is not relevant (the
cut-off is not needed). Using the truncated power law is "more" general, and better suits a
variety of larger scale problems.

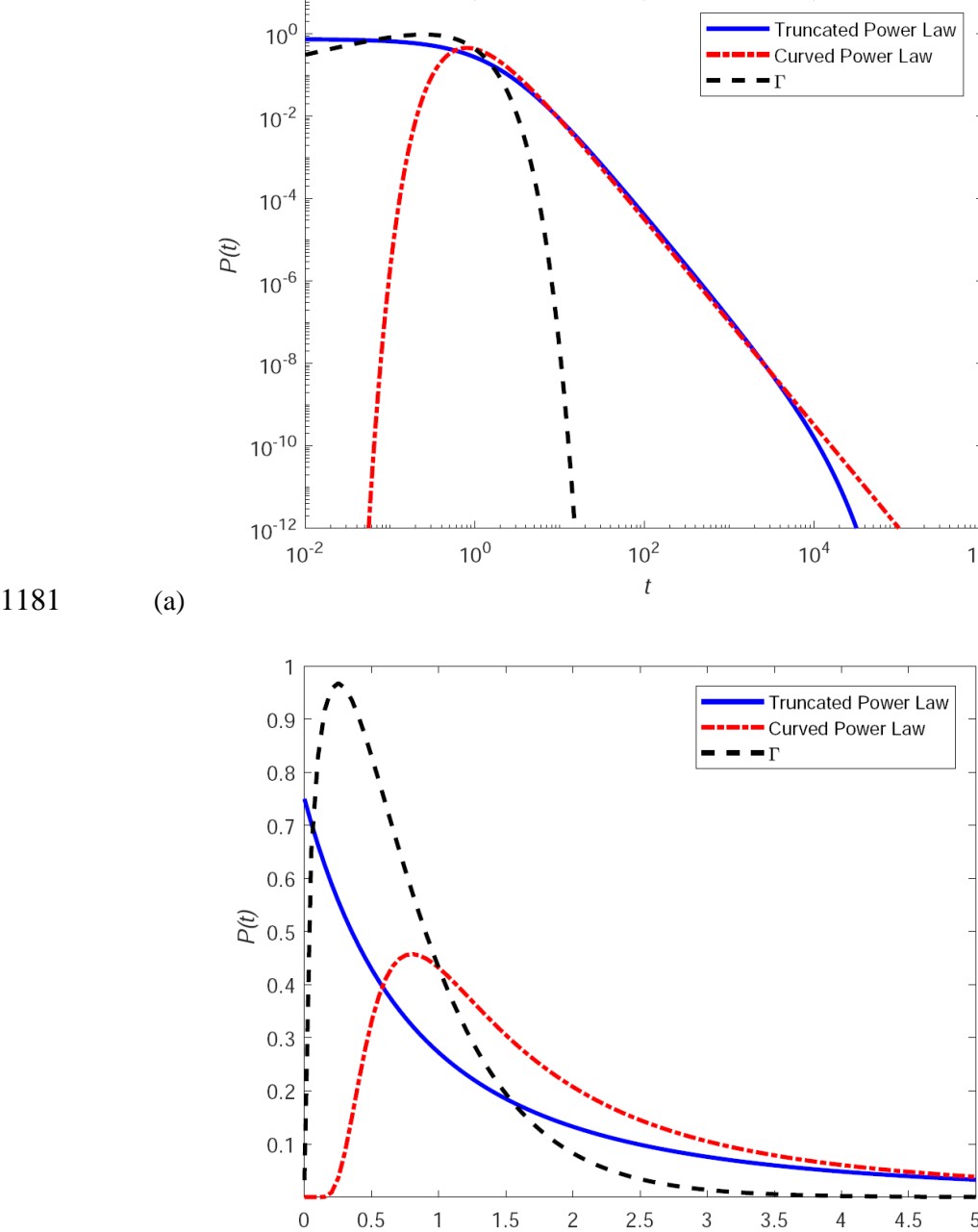

(a)
(b)
**Figure 7** Truncated power law, curved power law (inverse gamma distribution) and gamma
distribution, for the specific parameters: $\beta = 1.5$, $t_1 = 1$, $t_2 = 10^3$, $t^* = t_1$. **(a)** Log-log scale to
emphasise the long-time tailing behaviour. **(b)** Linear scale.

We now consider a specific example that demonstrates the relevance and applicability of
the CTRW framework for chemical transport in surface water systems, keeping the above
arguments in mind. Referring to the 2D case shown in Fig. 6a, we consider the effective
(travel time distribution) response, $h(t)$, to a rainfall pulse containing a chemical species over
the entire area of a catchment. Every point over this area may considered a source of chemical
species ("tracer"). A stream running through the catchment acts as a line sink (collector) for
the tracer. This catchment picture can be idealised as two rectangles straddling this stream
sink (Fig. 6a). Measurements of tracer arrivals at a control point downstream of this stream
(known as an "absorbing boundary") yield a tracer arrival "counting rate" that is a
breakthrough curve.
The first-passage time distribution $F(\mathbf{l_s}, t)$ defines the travel time distribution from a
(pulse) source at the origin $\mathbf{l}$ to the point $\mathbf{l_s}$. Then the chemical tracer/species concentration at
position $\mathbf{l_s}$ and time $t$, $c_s(\mathbf{l_s}, t)$ is given by

$$c_s(\mathbf{l_s}, t) = \int_0^\infty \sum_{\mathbf{l} \in \Omega} F(\mathbf{l_s} - \mathbf{l}, t') c_R(\mathbf{l}, t - t') dt' \qquad \text{(Eq. 15)}$$

where $c_R(\mathbf{l}, t)$ is the chemical input from rainfall at a position $\mathbf{l}$ in a catchment of area $\Omega$.
Referring then to Fig. 6a, because we sample chemical arrivals downstream, we can consider
the sampling position as an "instantaneous" integration of all chemical species/tracer arrivals
from the catchment pathways along the entire length of the stream. Travel time within the
stream can generally be assumed negligible, relative to the catchment travel times, as stream
velocities are generally much faster than combined overland/subsurface flows. We thus
determine the total chemical flux into the stream by integrating over all chemical inputs in the
catchment that reach the stream; this defines overall first-passage time distributions at the
downstream measurement point. Assuming that all of the sampling positions in $\mathbf{l_s}$ are small
regions compared to $\Omega$, then $c_s(\mathbf{l_s}, t) \approx c_s$. For uniform rainfall distribution over $\Omega$, we have
$c_R(\mathbf{l}, t) \approx c_R(t)$, and we can hence define for the effective, overall response (travel time
distribution)

$$h(t) \equiv \sum_{\mathbf{l} \in \Omega} F(\mathbf{l}, t) \ . \qquad \text{(Eq. 16)}$$

Long-term measurements of chloride tracer concentrations $c_R(t)$ in the rainfall over a
catchment area in Plynlimon, Wales, were compared to the time series of the chloride tracer
concentration $c_s(t)$ in the catchment Hafren stream (Kirchner et al., 2000). These authors
related the input and output concentrations through the convolution integral

$$c_s(t) = \int_0^\infty h(t') \, c_R(t - t') dt' \ . \qquad \text{(Eq. 17)}$$

Using a spectral analysis, Kirchner et al. (2000) concluded that overall chloride transport in
the catchment scaled as $h(t) \sim t^{-m}$, with $m \approx 0.5$, over a time period from 0.01 to 10 years.
They reported similar scaling in North American and Scandinavian field sites with $m \approx 0.4$–
1228 0.65.
Kirchner et al. (2000) continued their analysis by noting (i) that an exponential travel time
distribution (which is implicit in the advection-dispersion equation; see discussion above Eq.
(11)) does not match the data, and (ii) that conceptualization of the entire catchment as a
single flow path, and use of the advection-dispersion equation to describe travel times, do not
correctly match even the basic character of the chloride concentration arrivals. The authors
concluded that catchment travel time distributions should be quantified as an approximate
power law distribution, to correctly account for long-time chemical retention and release in
catchments, and defined $h(t)$ as a gamma distribution (recall Eq. (14)). It should be recognised
that the choice of a gamma distribution is empirical, and other functions can generate similar
behaviours in the spectral (Laplace or Fourier) domain. Significantly, the slope identified by
Kirchner et al. (2000) reflects high frequencies, i.e., short time scales; several decades of
tracer data to validate the power spectrum at low frequencies were not available.
Scher et al. (2002) reanalysed this catchment system behaviour with the CTRW
framework, arguing that subsurface flow and transport are dominant factors controlling the
overall chemical species arrival to the stream outlet measurement point. Based on Eqs. (15)
and (16), they first (re)examined the solution of the one-dimensional advection-dispersion
equation; they confirmed that the temporal dependence of $h(t)$ does not represent the field
measurements (similar to Kirchner et al., 2000). Significantly, though, they employed a pure
power law form of the transition time distribution, $\psi(t) \sim t^{-1-\beta}$, and developed Eqs. (15) and
(16) – based on the seminal analysis of Scher and Montroll (1975) – to obtain

$$h(t) \sim \begin{cases} t^{-1+\beta}, & t < t^* \\ t^{-1-\beta}, & t > t^* \end{cases} \qquad \text{for } 0 < \beta < 1. \qquad \text{(Eq. 18)}$$


The turnover time $t^*$ between these two slopes arises naturally as an outcome of chemical
transport in the system embodied in Eq. (16). The smaller times represent chemical inputs
following along fastest flow paths to the sampling point; for $t > t^*$, all chemical inputs over
the entire catchment area are contributing particles to the sampling point, as accounted for in
Eq. (17). In this latter case, the power law represents the overall particle movement in the
domain, but especially the effects of the slow particles (longer transition times and influence
of less mobile zones) and the longer travel distances.
In the context of the Hafren stream system, the turnover time $t^*$ was estimated as about 10
years (Scher et al., 2002), in agreement with findings and measurement range of Kirchner et
al. (2000), with $\beta = 0.5$. Figure 8 shows a representative plot of Eq. (18) for this system. As
noted in Scher et al. (2002), it remains to analyse measurements to confirm the turnover to the
longer-time $t^{-1-\beta}$ scaling behaviour, which is indicative of extremely long retention times.
Note that high-resolution measurements of low concentration levels in water are generally
required to analyse these longer-time tails. The key recognition here is that while the effective
catchment response *may potentially*, initially (i.e, at relatively short times), be represented by
a type of gamma distribution (i.e., a power law $\sim t^{-1+\beta}$, ignoring the exponential cut-off) at
sufficiently small times (<10 years in the case of the Hafren catchment) – and this is
embodied in the CTRW framework as seen in Eq. (18) – full (CTRW framework) power law
behaviour (i.e., $\sim t^{-1-\beta}$) over longer times should also be incorporated to describe expected
long-term catchment retention behaviour. An evolution to Fickian transport, via an
exponential cut-off at very long times, can also be included (if relevant). To conclude, while
direct, quantitative application of CTRW to analysis of chemical transport at the catchment
scale remains to be done, it appears – on the basis of the conceptual pictures, extensive
application to subsurface systems and direct similarities to catchment systems, and the robust
and general nature of the CTRW formulation – to be a highly promising avenue for future
research.

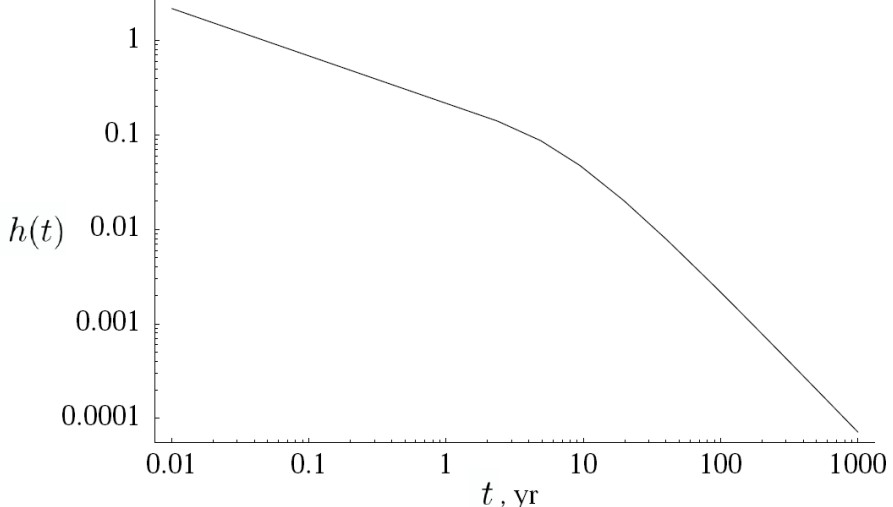

**Figure 8** A log-log plot of $h(t)$ vs. $t$ (after Scher et al., 2002; Copyright 2002, with permission
from the American Geophysical Union).

## 4   CONCLUSIONS AND PERSPECTIVES

### 4.1   Preferential flow and non-Gaussian travel times: The spatial and the temporal manifestation of organized complexity

Based on Sects. 2 and 3, we can state that (a) preferential flow and related non-Fickian transport is an omnipresent, unifying element between both water worlds, and (b) the CTRW framework can effectively quantify and predict non-Fickian transport of water and chemicals species in a manner that connects to and clearly steps beyond the advection-dispersion paradigm. In this section, we link these insights to our central proposition that preferential flow is a prime manifestation of how a local-scale heterogeneous flow process causes a macroscale organised flow pattern in *space*. The key is to acknowledge that organisation manifests also through organised dynamic behaviour in *time*, which occurs through non-Fickian travel time distributions of water and chemical species. Note that the degree of organisation in *space* manifests in the deviation of spatial patterns of system characteristics or fluid flow from the maximum entropy pattern. The latter corresponds, in the case that the mean value is known, to a uniform distribution of system characteristics and/or a uniform flow pattern. Along the same lines, we propose that the degree of organisation in dynamic behaviour in *time* manifests through the deviation of the breakthrough curve from the case of a well-mixed Gaussian system, which is quantified within the CTRW framework based on the power law exponent. A power law exponent ≥2 corresponds to well a mixed travel time distribution. The latter reflect a spatial concentration equal to a Gaussian, which maximises entropy when the *mean* and the *variance* are known (Trefry et al., 2003).

In terms of how power law transition distributions are linked to the formation, evolution and function of preferential flow paths in surface water systems, and how and if they can be expected to improve representation thereof in models, we first emphasise that power law transition time distributions are linked to the *function* of preferential flow paths, but not to

their formation and evolution. It is clear and well-known that preferential flow implies non-
Fickian residence times or travel distance. But what has not been recognized, though, is that
the fingerprint of preferential flow in the overall travel time distribution can be captured by a
(truncated) power law for the transition time distribution; and through the related exponent we
can quantify the deviation from the well-mixed Fickian case. As discussed in Sect. 2.4, the
findings of Edery et al. (2014) suggest a further connection between the characteristics of an
aquifer and the power law exponent in breakthrough curves. This implies that the fitted
parameters are a macroscale fingerprint of spatial media characteristics that determine the
temporal arrival of chemical species. While we do not expect that this relation is unique, it
does imply that "fitted" parameters have a physical meaning that can be used to constrain
characteristics of the domain (i.e., the hydrological landscape mentioned above) in a spatially
distributed model.
We argue that this should also hold for other complex media characteristics that relate to
their spatial organization, such as the correlation length or topology of preferential flow paths.
We therefore suggest that these insights offer opportunities to relate signatures of spatial
organization in flow patterns to signatures of temporal organization in breakthrough curves.
For both perspectives, we can quantify organization using information entropy, as we showed
in Sect. 2.4. These arguments might also offer, ultimately, opportunities to test whether
hydrological systems and their preferential flow networks co-evolve towards more energy
efficient drainage, which can also be quantified (Kleidon et al. 2013; Zehe et al., 2019;
Savenije and Hrachowitz, 2017). We leave a more detailed reflection on this for future
studies.

## 4.2   Overall conclusions and perspectives

In an effort to integrate and unify conceptualisation and quantitative modelling of the two
"water worlds"– surface water and groundwater systems – we recognise preferential fluid
flows as a unifying element and consider them as a manifestation of self-organisation.
Preferential flows hinder perfect mixing within a system, due to a more "energy efficient" and
hence faster throughput of water, which affects residence times of water, matter and chemical
species in hydrological systems across all scales. While our main focus here is on the role of
preferential flow for residence times and chemical transport, we relate our proposed unifying
concept to role of preferential flow for energy conversions and energy dissipation associated
with flows of water and mass.
Essentially, we have proposed that related conceptualisations on the role of heterogeneity
and preferential fluid flow for chemical species transport, and its quantitative characterisation,
can be unified in terms of a theory, based on the CTRW framework, that connects these two
water worlds in a dynamic framework. We emphasise the occurrence of power law
behaviours that characterise travel times of chemical species, and highlight the critical role
played by system heterogeneity and chemical species residence times, which are distinct from
travel times of water. In particular, we compare and contrast specific power law distributions,
and argue that the closely related inverse gamma and algebraic power law distributions are
more appropriate than the oft-used gamma distribution to quantify chemical species transport.
Moreover, we identify deviations from well-mixed Gaussian transport as a manifestation
of self-organised dynamic behaviour in time, and the power law exponent as a suitable means

to measure the strength of this deviation. Along a complementary line, we propose that self-organisation in space is immanent primarily through strongly localised preferential flow through rill and river networks at the land surface. We relate the degree of spatial organisation to the deviation of the flow pattern from spatially homogeneous flow, which is a state of maximum entropy. In this context, we reflect on the ongoing controversial discussion regarding whether or not self-organisation in open hydrological systems leads to evolution to a more energy efficient or even thermodynamic optimal system configuration. Finally, we propose that our concept of temporally organised travel times can help to test the possible emergence of thermodynamic optimality. Complementary to this idea, we suggest that an energetic perspective of chemical species transport may help to explain the organisation of travel paths (Fig. 4), in the sense that contrary to common assumptions, preferential pathways often include "bottlenecks" of low hydraulic conductivity. A testable option could be that chemical species travel along the path of maximum power, with power being defined in this case as flow of chemical energy (rather than flow of kinetic energy) through the system.

Overall, we conclude that self-organisation arises equally in surface water and groundwater systems, as local heterogeneity and disorder in fluid flow and chemical transport processes lead to ordered behaviour at the macroscale. Naturally, the surface water community has developed a strong emphasis on the *localised spatial fingerprints*, because rills and rivers are clearly visible on land (Fig. 1), while the groundwater community has focused more naturally on *non-local temporal fingerprints*, as the flow paths are largely unobservable. But these are just two sides of the same conceptual picture of organised complexity (Dooge, 1986).

**Competing interests.** The authors declare that they have no conflict of interest.

**Acknowledgements.** The authors thank Markus Hrachowitz, Nicolas Rodriguez, Matthias Sprenger, and an anonymous referee for particularly constructive reviews of this work. B.B. gratefully acknowledges the support of research grants from the Israel Water Authority (Grant No. 45015199895) and the Israel Science Foundation (Grant No. 485/16); he thanks Harvey Scher for in-depth discussions. B.B. holds the Sam Zuckerberg Professorial Chair in Hydrology. E.Z. gratefully acknowledges intellectual support by the "Catchments as Organized Systems" (CAOS) research unit and funding of the German Research Foundation, DFG, (FOR 1598, ZE 533/11-1, ZE 533/12-1).

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
