# Peer review of "Surface water and groundwater: Unifying conceptualization and quantification of the two "water worlds""

_Hydrology and Earth System Sciences, 2019_

## Referee Comment (RC1) · Markus Hrachowitz (Referee) · 7 Nov 2019

In the manuscript "Surface water and groundwater: unifying conceptualization and quantification of the wo water worlds" by Berkowitz and Zehe, the authors aim to raise awareness for the unjustified and unnecessary distinction between surface and groundwater hydrology. They make the case that both systems are controlled by largely the same principles – one of these principles being the energetically necessary development of preferential flow paths.

The manuscript touches a highly interesting topic and it has, without doubt, the potential to become an important reference for future studies. Although I really like and

appreciate the overall intention and objective of the manuscript, I also feel that the manuscript could strongly benefit from the authors taking a step back to re-reflect some parts of their work.

(1) Throughout the manuscript it remains somewhat unclear what the authors want to convey. While some parts of the text read like a very interesting, yet general reflection on the structural similarity of surface and groundwater systems and the general principles behind that, other parts are very technical descriptions of one specific aspect (i.e. power law transition distributions) of groundwater and potentially surface water systems. Both parts are very interesting, but there is little clear coherence between them in the text. It therefore remains somewhat vague, how and if power law transition distributions are in detail linked to the formation, evolution and function of preferential flow paths in surface water systems and how and if they can be expected to improve representation thereof in models.

(2) Linked to comment (1), it is difficult to discern from the text what the original and novel contribution of this manuscript is. In other words, it remains unclear if the manuscript is intended to provide a review of the state-of-the-art together with guidance towards promising future research avenues or if the thermodynamic considerations and the proposed power law transition distributions are a novel development that is introduced here for the first time (which I do not suspect). I therefore strongly encourage the authors to more clearly distinguish between existing literature and potential novel contributions.

(3) I strongly agree with the authors that eventually the surface- and groundwater communities need to converge towards "unified" conceptualizations – we are talking about the same physical system after all. However, and in contrast to the authors, I believe that different modelling approaches are not mutually exclusive. Rather, they should all be embraced and exploited to their fullest to learn about the system. In other words, I think, different modelling approaches are complementary in what can be learned from them. Here, the authors provide a beautiful example of how theoretical considerations

and "physically-based" models can teach us about real world systems. Their findings that power law transition distributions may be more suitable than other, currently used distributions can be of considerable interest for other modelling approaches. I think it would therefore be very helpful to emphasize this complementary aspect. Now large parts of the manuscript read like as if typical top-down model approaches cannot deal with the celerity-velocity dualism, nor that these models could resolve the incomplete mixing. I strongly disagree with this notion. While it is true that simple, lumped convolution integral approaches have limited use, they are quite outdated and cannot be seen as state-of-the-art anymore, for reasons also highlighted by the authors in this manuscript. That they are still widely used in the community is a different problem. On the other hand, the authors claim that conceptual multi-box models similarly, cannot represent the system nor reproduce its dynamics in terms of both, water and chemistry. I disagree with this opinion in the strongest terms, as there are many papers, in particular over the last 5-10 years, in which the opposite was shown. I agree that standard conceptual box models cannot simultaneously reproduce water flows and solute concentrations (and thus water/solute age distributions). However, as already shown almost 20 years ago by Seibert et al. (2003; HP), the addition of "mixing" assumptions and hydraulically "passive" mixing volumes has demonstrated great ability in resolving this problem (and thereby the "old water paradox" – which is not a paradox anyway, really). Of course, many of these papers (e.g. Shaw et al., 2007, JoH; Fenicia et al., 2008, WRR; Birkel et al., 2010, HP; Dunn et al., 2007, WRR; McMillan et al., 2012, WRR) started with the assumption of complete mixing in the individual reservoirs. This was done not out of conviction but rather in absence of more detailed information. However, soon it was realized that complete mixing in the presence of preferential flowpaths is an unsuitable assumption. Subsequent work therefore adopted the use of incomplete mixing at least for the unsaturated root-zone, and which could either be time-invariant (e.g. Hrachowitz et al., 2015, HP) or time-varying as a function of wetness conditions (e.g. Hrachowitz et al., 2013; HESS). In a parallel development, similar considerations were made using the SAS-function approach (e.g. van der Velde et al.,

2010,2012; WRR; Benettin et al., 2015a,b, WRR; Harman, 2015; WRR; Rinaldo et al., 2015, WRR and many others). Here, please note that the SAS-function approach is functionally and even mathematically (!!) equivalent to the mixing coefficient approach above, as described in detail in Hrachowitz et al. (2016; Wires). The only difference is in the choice of the SAS-function (piece-wise linear for mixing coefficient approach and typically gamma or beta distributions for SAS approach) and in the way the involved processes are described semantically. It is true that many applications(!) of the SAS-function approach limit themselves to overly simplified representations of hydrological systems. However, this does not mean that the general concept behind it is invalid. The application of different individual(!) mixing coefficients (and thus SAS-functions) for different individual(!) system components has, on the contrary, already proven very useful (Fenicia et al., 2010, WRR; Hrachowitz et al. 2013, HESS; Hrachowitz et al. 2015, HP; Hrachowitz et al. 2016, Wires). With the new information provided in this manuscript, a logical next step should thus be to check if these types of models can reproduce power law transit time distributions and if power law distributions would be suitable as SAS-functions in different components of the system (e.g. in the unsaturated root zone for drainage and evaporation; in groundwater for drainage). In fact in our paper Hrachowitz et al. (2013; HESS), using incomplete, temporally varying mixing in the unsaturated zone, we found that the system overall transit time distributions (i.e. TTD of the modelled combined outflows) had longer than gamma distribution tails. An observation that we could not make too much sense of at that time. However, it could fit very well into the reasoning presented by the authors here. in that sense, this would be a beautiful example of how different modelling approaches could learn from each other. It would make the manuscript much stronger if the authors emphasized how their findings could be helpful for other modelling approaches and if the authors invested some more effort in being more accurate in their description of what different model approaches can do. In the current description the authors seem to equate what "is" done with these models with what "can" be done. This is not a valid assumption. In fact we can do much more than is typically done.

Minor comments:

p.3,l.96: what are "uncorrelated noise pattern"?

p.4,l.133: Hrachowitz et al. (2016) would fit nicely here as reference

p.4,l.146: not sure why this is referred to as "weak interaction" here. Given its importance for flow velocities and sediment transport, should friction not cause a "strong interaction" between water and solids?

p.5,l.161 and elsewhere: why only chemicals? This is also true for the behaviour of individual water molecules. The difference is that we can normally not tag and distinguish them. But this does not make the general process valid for chemicals only. There is also imperfect mixing of waters of different ages/provenance. Please try to be more precise in these formulations. In addition, please note that this is exactly what incomplete mixing assumptions and/or non-uniform SAS-functions try to mimic.

p.5,l.178: also depending on the pre-storm wetness conditions

p.6,l.203-204: I disagree with this statement. While it is true that conceptual models are often applied in a careless way, there is a wide body of literature that describes the necessity and value of treating models as hypotheses that need to be tested (e.g. Fenicia et al., 2014; Kavetski et al., 2011; Clark et al., 2011;2015), which, in an iterative approach, allows us to learn about the system. In addition, in Hrachowitz and Clark (2017) we argue that ideally, these catchment-scale, effective parameters should reflect real world numbers. The actual challenge is to find these numbers, which is far from trivial for many parameters, but which was shown to be feasible for others, e.g. Master-Recession Curve (Lamb and Beven, 1997) or the storage capacity in the unsaturated root zone (e.g. deBoer-Euser et al., 2016; Nijzink et al., 2016)

p.6,l.210: I also disagree here. these relations remain specific for the spectrum of environments they have been developed for. If they relations were developed using a wide range of different landscapes, as done for example for the determination of global

parameters in the mhM model (e.g. Samaniego et al., 2010) than they will also give us a more general picture.

p.6,l.219: No, I strongly disagree! Why should a parameter that describes the aggregated effects of heterogeneity be an abstract quantity? I think it rather provides the macroscale perspective and it can in some instances already be independently observed/estimated (see comment above about Master-Recession-Curve and storage capacity in root zone)

p.6,l.223-224: no, although there is without doubt some tendency to believe that parameter search can help, I do not share this. I rather think that, for all the uncertainties involved, we need to start with step-by-step limiting the feasible parameter space by identifying and eliminating solutions that are inconsistent with our data and our understanding of the system. That is essentially the opposite of finding the "optimal" parameter set and relatively independent of parameter search algorithms.

p.6,l.231-234: with all appreciation for the authors, but this is too bold a statement, which I need to consider as plainly wrong. First, these models *can* resolve the celerity/velocity dualism, when implemented as described above (mixing coefficients/SAS-function). And this, second, has already been demonstrated with a plethora of articles on combined modelling of water flow and (non-)conservative solute concentrations (e.g. chloride, nitrate, silica, ANC, EC, DOC and many others; see above references and references therein)

p.7,l.247-263: while Darcy-Richards does indeed, and probably rightfully so, receive criticism, I think these problems can be fixed within the near future (as also suggested by the list of improvements given here by the authors). I rather consider another point why these models receive much criticism: the fact that the sheer number of parameters needed can never be observed at the spatial resolution and scale(!!) of interest. Either we then need to use our scarce, existing observations to inform our model, in which case we will upscale homogeneity (as the spatial correlation fields of our system properties are unknown!), or we will need to calibrate these models, in which case we will obviously run into the problem of equifinality and our inability to meaningfully constrain our models.

p.7,l.264-265: why "alternative"? should these tracers not also be very informative and helpful to implement and test the above models?

p.7,l.264-270: true, but mostly outdated, as these approaches are too simplistic.

p.7,l.271-283: in this paragraph the authors seem to be confusing different approaches. At least it reads in a rather incoherent way. for example, Rinaldo et al. (2015) and Botter et al. (2011) describe the same general concept. Further, the system overall SAS-function can also emerge from the choice of SAS functions from individual components, which can be a calibration parameter or informed by observations/theoretical considerations as demonstrated by the authors of this manuscript. Finally, yes, the gamma distribution is often used, but as often other choices, such as the beta-distribution (e.g. van der Velde et al., 2012) or some piece-wise linear distributions are made (Hrachowitz et al., 2013,2015). In any case, and as mentioned above, these choices are not necessarily made out of conviction but rather due to a lack of more information and can be easily adjusted. In particular, it will be very help to get a better understanding how the tails of the distributions should look like.

p.10,l.364-370: unclear in how far this is different to what mixing coefficient/SAS-functions do.

p.10,l.375-378: I believe there is quite a good understanding in the community that there is no binary distinction between separated pools of water. The "two water worlds" need to be understood, prosaic literature terms, as a hyperbole – an exaggerated analogy, i.e. a pointed description of a concept.

p.10,l.379-380: perhaps better "drainage and evaporative fluxes"

p.11,l.431 and elsewhere: I am not sure if the term "chemical transport" is the best

way to express what you want to express here. why not keeping it more general to "transport", which then indeed is further divided into physical, bio-physical and chemical components in the transport processes.

p.12,l.454: conservative transport?

p.14,l.500: where can I see this?

p.15,l.500-503: sure, but would be good to introduce what CTRW is before shown results.

p.15,l.550ff: as also demonstrated by Hrachowitz et al. (2015 – sorry to again bring in one of my own papers here) using a conceptual model with mixing coefficients, where we showed that conservative (!) chloride transport is slower than water transport (evapoconcentration)

p.17,l.594ff: I really like this analogies here, but a stronger explanation of and link between spatial and temporal heterogeneity needed. Why are they different? What is going on there?

p.17,l.632ff, section 3.3: in how far is this different to SAS? This does not clearly come out here. I think it would be very interesting for the reader to clarify this. Is there a fundamental difference or is it only the choice of the transition time distributions?

p.17,l.646 and elsewhere: I noted the use of "transition" times instead of what is standard use in surface water hydrology: "transit" times. Was this made on purpose? Is there a difference? If yes, please specify. If no, please also clarify.

p.17,l.670: water molecules, too, are subject to both! It is just that its more difficult for us to observe

p.21,l.802-804: because that was the best information we had so far.

p.22,l.821: not sure this is correct "...sum of a gamma function(which is also a gamma function)...". Please check if this makes sense. Do you mean the scaled incomplete

gamma function? Should it not rather reader something like:"the sum of 2 random variables that follow a gamma distribution can also be described by a gamma distribution"?

p.22,l.824: well, no. Not if suitable local (i.e. individual for each system component) mixing coefficients/SAS-functions are chosen.

p.22,l.839-841: I am not sure that that reference is a good example. First, the NSE is a very pessimistic performance metric in cases there is a low signal-to-noise ratio in the target variable (which is the fact in the Weiherbach: the stream isotope composition is strongly damped and essentially plots close to a straight line – any small deviation from that – error or real process – causes strong effects on NSE). Second, the choice of model also only allows a rather rudimentary partitioning and routing of water fluxes, which will have a considerable effect on the tracer composition.

p.23,l.868-873: it is difficult to judge for the reader if the CTRW framework is more physically justified than other models. What also remains unclear: is CTRW necessary to model long tails? Or can suitable distributions be used in other (conceptual) models to reproduce similar results?

Best wishes, Markus Hrachowitz

References:

Benettin, P., Kirchner, J. W., Rinaldo, A., & Botter, G. (2015a). Modeling chloride transport using travel time distributions at Plynlimon, Wales. Water Resources Research, 51(5), 3259-3276.

Benettin, P., Bailey, S. W., Campbell, J. L., Green, M. B., Rinaldo, A., Likens, G. E., ... & Botter, G. (2015b). Linking water age and solute dynamics in streamflow at the Hubbard Brook Experimental Forest, NH, USA. Water Resources Research, 51(11), 9256-9272.

Birkel, C., Tetzlaff, D., Dunn, S. M., & Soulsby, C. (2010). Towards a simple dynamic process conceptualization in rainfall–runoff models using multi‐criteria calibration

and tracers in temperate, upland catchments. Hydrological Processes: An International Journal, 24(3), 260-275.

Clark, M. P., Kavetski, D., & Fenicia, F. (2011). Pursuing the method of multiple working hypotheses for hydrological modeling. Water Resources Research, 47(9).

Clark, M. P., Fan, Y., Lawrence, D. M., Adam, J. C., Bolster, D., Gochis, D. J., ... & Maxwell, R. M. (2015). Improving the representation of hydrologic processes in Earth System Models. Water Resources Research, 51(8), 5929-5956.

de Boer‐Euser, T., McMillan, H. K., Hrachowitz, M., Winsemius, H. C., & Savenije, H. H. (2016). Influence of soil and climate on root zone storage capacity. Water Resources Research, 52(3), 2009-2024.

Dunn, S. M., McDonnell, J. J., & Vaché, K. B. (2007). Factors influencing the residence time of catchment waters: A virtual experiment approach. Water Resources Research, 43(6).

Fenicia, F., McDonnell, J. J., & Savenije, H. H. (2008). Learning from model improvement: On the contribution of complementary data to process understanding. Water Resources Research, 44(6).

Fenicia, F., Wrede, S., Kavetski, D., Pfister, L., Hoffmann, L., Savenije, H. H., & McDonnell, J. J. (2010). Assessing the impact of mixing assumptions on the estimation of streamwater mean residence time. Hydrological Processes, 24(12), 1730-1741.

Fenicia, F., Kavetski, D., Savenije, H. H., Clark, M. P., Schoups, G., Pfister, L., & Freer, J. (2014). Catchment properties, function, and conceptual model representation: is there a correspondence?. Hydrological Processes, 28(4), 2451-2467.

Harman, C. J. (2015). Time‐variable transit time distributions and transport: Theory and application to storage‐dependent transport of chloride in a watershed. Water Resources Research, 51(1), 1-30.

[Figure]

Hrachowitz, M., & Clark, M. P. (2017). HESS Opinions: The complementary merits of competing modelling philosophies in hydrology. Hydrology and Earth System Sciences, 21(8), 3953-3973.

Hrachowitz, M., Savenije, H., Bogaard, T. A., Tetzlaff, D., & Soulsby, C. (2013). What can flux tracking teach us about water age distribution patterns and their temporal dynamics?. Hydrology and Earth System Sciences 17, 533-564.

Hrachowitz, M., Fovet, O., Ruiz, L., & Savenije, H. H. (2015). Transit time distributions, legacy contamination and variability in biogeochemical $1/f\alpha$ scaling: how are hydrological response dynamics linked to water quality at the catchment scale?. Hydrological Processes, 29(25), 5241-5256.

Hrachowitz, M., Benettin, P., Van Breukelen, B. M., Fovet, O., Howden, N. J., Ruiz, L., ... & Wade, A. J. (2016). Transit times—the link between hydrology and water quality at the catchment scale. Wiley Interdisciplinary Reviews: Water, 3(5), 629-657.

Kavetski, D., & Fenicia, F. (2011). Elements of a flexible approach for conceptual hydrological modeling: 2. Application and experimental insights. Water Resources Research, 47(11).

Lamb, R., & Beven, K. (1997). Using interactive recession curve analysis to specify a general catchment storage model. Hydrology and Earth System Sciences, 1(1), 101-113.

McMillan, H., Tetzlaff, D., Clark, M., & Soulsby, C. (2012). Do time‐variable tracers aid the evaluation of hydrological model structure? A multimodel approach. Water Resources Research, 48(5).

Nijzink, R., Hutton, C., Pechlivanidis, I., Capell, R., Arheimer, B., Freer, J., ... & Hrachowitz, M. (2016). The evolution of root-zone moisture capacities after deforestation: a step towards hydrological predictions under change?. Hydrology and Earth System Sciences, 20(12), 4775-4799.

Rinaldo, A., Benettin, P., Harman, C. J., Hrachowitz, M., McGuire, K. J., Van Der Velde, Y., ... & Botter, G. (2015). Storage selection functions: A coherent framework for quantifying how catchments store and release water and solutes. Water Resources Research, 51(6), 4840-4847.

Samaniego, L., Kumar, R., & Attinger, S. (2010). Multiscale parameter regionalization of a grid‐based hydrologic model at the mesoscale. Water Resources Research, 46(5).

Shaw, S. B., Harpold, A. A., Taylor, J. C., & Walter, M. T. (2008). Investigating a high resolution, stream chloride time series from the Biscuit Brook catchment, Catskills, NY. Journal of hydrology, 348(3-4), 245-256.

Van Der Velde, Y., Torfs, P. J. J. F., Van Der Zee, S. E. A. T. M., & Uijlenhoet, R. (2012). Quantifying catchment‐scale mixing and its effect on time‐varying travel time distributions. Water Resources Research, 48(6).

───────────────────────────────

---

## Author Comment (AC1) · 23 Nov 2019

We sincerely thank Markus Hrachowitz (MH) for his detailed, constructive and very thoughtful assessment of our manuscript. We are confident that we can fully address his valuable recommendations in the revised manuscript, as detailed in our response to the individual points.

**MH:** In the manuscript "Surface water and groundwater: unifying conceptualization and quantification of the two water worlds" by Berkowitz and Zehe, the authors aim to raise awareness for the unjustified and unnecessary distinction between surface and groundwater hydrology. They make the case that both systems are controlled by

largely the same principles – one of these principles being the energetically necessary development of preferential flow paths. The manuscript touches a highly interesting topic and it has, without doubt, the potential to become an important reference for future studies. Although I really like and appreciate the overall intention and objective of the manuscript, I also feel that the manuscript could strongly benefit from the authors taking a step back to re-reflect some parts of their work.

**RESPONSE:** We thank MH for constructive comments. In our revised manuscript, we will certainly modify the text to clarify the arguments and discussion, along the lines suggested by MH in the specific comments below. To expand on and re-emphasize the "philosophy" of our manuscript, we note the following: Scientific communication and cooperation between the two communities is generally very limited, largely because of perceptions that the research problems and natures of the system dynamics are distinctly different and even unrelated. Indeed, current theoretical frameworks and experimental methods to characterize, measure, quantify, and model fluid flow - and related transport of chemicals and energy - are generally distinct and separate. And yet, as these "worlds" are two perspectives of the same terrestrial hydrological system, we propose that their dynamics are governed by common laws and principles. Moreover, each community has developed sophisticated modelling and measurement capabilities - which have led to significant scientific advances over the last two decades - that could benefit the other community and address its outstanding, unsolved problems. We synthesize the methods and thinking of the two communities in an effort to spark sharing and integration of future research efforts.

**MH:** (1) Throughout the manuscript it remains somewhat unclear what the authors want to convey. While some parts of the text read like a very interesting, yet general reflection on the structural similarity of surface and groundwater systems and the general principles behind that, other parts are very technical descriptions of one specific aspect (i.e. power law transition distributions) of groundwater and potentially surface water systems. Both parts are very interesting, but there is little clear coherence between them in the text. It therefore remains somewhat vague, how and if power law
transition distributions are in detail linked to the formation, evolution and function of preferential flow paths in surface water systems and how and if they can be expected to improve representation thereof in models.

**RESPONSE:** We agree that two these different parts of the manuscript differ with respect to level mathematical detail. In fact, we considered at length various ways to address mathematical aspects, and reflected on the whether or not to include mathematical expressions as done in section 3. At the time, we were concerned that this might make the manuscript too dense. However, in light of MH's comments, we will introduce several key revisions: (1) We will clarify in the Introduction that our manuscript synthesizes concepts and methods from the generally disparate surface water (catchment hydrology) and groundwater research communities, and is in this sense innovative. We are not simply reviewing material or offering an opinion. (2) In the context of (1), we focus in section 2 on showing - in phenomenological and conceptual terms - how the structure and patterning of fluid flow (and chemical species) in surface water and groundwater systems are similar, and thus amenable to similar quantitative treatments. In the revised manuscript, we will add basic equations relevant to the discussion, to offer a firmer quantitative background to the general principles we discuss. We will then clarify that section 3 (i) offers a first effort at defining specific conceptual and quantitative tools, (ii) introduces the CTRW framework in this concept, with a clear connection to microscale physics, and places it in perspective relative to the well-known advection-dispersion equation (so that presentation of 6, and (actually 5) basic equations is necessary), (iii) offers a new development and insight, in terms of comparing and contrasting power law and inverse Gamma distributions (groundwater literature; more appropriate to describe long tailing in residence time distributions) with Gamma distributions (surface water literature), which includes 2 additional equations, and (iv) uses the preceding discussion and insights to show (with 4 additional equations) how surface water systems (catchment response to chemical transport) can be treated within the CTRW framework. We emphasize that we must retain essential mathematics, but will ensure that as the manuscript aims to bridge two communities, it

**HESSD**
has to be in a style that it is attractive for both audiences. (3) The reviewer asks how power law transition distributions are linked to the formation, evolution and function of preferential flow paths in surface water systems and how and if they can be expected to improve representation thereof in models. We clarify that power law transition time distributions are linked to the "function" of preferential flow paths, but formation and evolution of (surface water) preferential flow paths are a separate matter. To add more: A key point of the more general part of the manuscript, that we shall stress even more, is that we step clearly beyond a mere statement of what does not happen if preferential flow occurs (in the sense that preferential flow implies non-Gaussian residence times or travel distance; we have known this for a long time) — we provide a statement of what does happen. The fingerprint of preferential flow in the travel time distribution can be captured by a truncated power law (for the transition time distribution, which underlies the overall travel time distribution); and through the related exponent we can quantify the deviation from the well mixed Gaussian case. The work of Edery et al. (2014), we present in section 2.3 suggests a further connection between the characteristics of the aquifer and the power law exponent in breakthrough curves. We argue that this should also hold for other complex media characteristics that relate to their spatial organization (such as the correlation length or topology of preferential flow paths). We are therefore suggesting that these insights offer opportunities to relate signatures of spatial organization in flow patterns to signatures of temporal organization in the breakthrough curves. For both perspectives, we can quantify organization using information entropy, as we will show in the revised manuscript. A specific example calculation will be included in section 2.3, based on Figure 3, to illustrate this point. These arguments also offer, ultimately, opportunities to test whether hydrological systems and their preferential flow networks co-evolve towards more energy efficient drainage, which can also be quantified.

**MH:** (2) Linked to comment (1), it is difficult to discern from the text what the original and novel contribution of this manuscript is. In other words, it remains unclear if the manuscript is intended to provide a review of the state-of-the-art together with guidance

**HESSD**
towards promising future research avenues or if the thermodynamic considerations and the proposed power law transition distributions are a novel development that is introduced here for the first time (which I do not suspect). I therefore strongly encourage the authors to more clearly distinguish between existing literature and potential novel **RESPONSE:** Thank you for this very valid point. As explained in our response to comment (1), we will indeed better explain that the manuscript relies largely on the synthesis of work that has already published in both communities, and proposes a common conceptual framework (we avoid the term theory, as this would require even more mathematics). We will also more clearly point out the completely new and novel results (e.g., the comparison of power law and inverse Gamma distributions against Gamma distributions; the quantification of organization in preferential flow patterns using information entropy). The latter calculation was presented by one of the authors during the recent 2019 EGU Leonardo conference in Luxembourg (Zehe et al., 2019).

**MH:** (3) I strongly agree with the authors that eventually the surface- and groundwater communities need to converge towards "unified" conceptualizations – we are talking about the same physical system after all. However, and in contrast to the authors, I believe that different modelling approaches are not mutually exclusive. Rather, they should all be embraced and exploited to their fullest to learn about the system. In other words, I think, different modelling approaches are complementary in what can be learned from them. Here, the authors provide a beautiful example of how theoretical considerations and "physically-based" models can teach us about real world systems. Their findings that power law transition distributions may be more suitable than other, currently used distributions can be of considerable interest for other modelling approaches. I think it would therefore be very helpful to emphasize this complementary aspect.

**RESPONSE:** We really thank MH for pinpointing this and for the detailed thoughts below. It was not our purpose to return to the old and very unfruitful debate about "which model paradigm is better", the conceptual one or the physically based one. So we will definitely revise this section (particularly stressing the valuable and highly relevant

**HESSD**
studies MH listed in his review) which corroborate that – regardless of the model type – modelling of chemical species transport must overcome the limitation of well-mixed assumptions. This can be achieved either as MH explained, using skewed transit time distributions (as the Gamma or inverse Gamma function), or the concept of SAS in conjunction with conceptual models for the mass balance. This can be also be done by using Darcy-Richards model in combination with particle tracking, and solving them in heterogeneous domains (as shown, e.g., by Edery et al. (2014)), and in partially saturated soils (e.g., Klaus and Zehe, 2010, 2011;Wienhoefer and Zehe, 2014) that represent preferential flow paths as connected, highly conductive pathways with close-to-zero retention properties.

**MH:** Now large parts of the manuscript read like as if typical top-down model approaches cannot deal with the celerity-velocity dualism, nor that these models could resolve the incomplete mixing. I strongly disagree with this notion. While it is true that simple, lumped convolution integral approaches have limited use, they are quite outdated and cannot be seen as state-of-the-art anymore, for reasons also highlighted by the authors in this manuscript. That they are still widely used in the community is a different problem. On the other hand, the authors claim that conceptual multi-box models similarly, cannot represent the system nor reproduce its dynamics in terms of both, water and chemistry. I disagree with this opinion in the strongest terms, as there are many papers, in particular over the last 5-10 years, in which the opposite was shown. I agree that standard conceptual box models cannot simultaneously reproduce water flows and solute concentrations (and thus water/solute age distributions).

However, as already shown almost 20 years ago by Seibert et al. (2003; HP), the addition of "mixing" assumptions and hydraulically "passive" mixing volumes has demonstrated great ability in resolving this problem (and thereby the "old water paradox" – which is not a paradox anyway, really). Of course, many of these papers (e.g. Shaw et al., 2007, JoH; Fenicia et al., 2008, WRR; Birkel et al., 2010, HP; Dunn et al., 2007, WRR; McMillan et al., 2012,) WRR) started with the assumption of complete mixing in
the individual reservoirs. This was done not out of conviction but rather in absence of more detailed information.

However, soon it was realized that complete mixing in the presence of preferential flow paths is an unsuitable assumption. Subsequent work therefore adopted the use of incomplete mixing at least for the unsaturated root-zone, and which could either be time-invariant (e.g. Hrachowitz et al., 2015, HP) or time-varying as a function of wetness conditions (e.g. Hrachowitz et al., 2013; HESS). In a parallel development, similar considerations were made using the SAS-function approach (e.g. van der Velde et al., 2010,2012; WRR; Benettin et al., 2015a,b, WRR; Harman, 2015; WRR; Rinaldo et al., 2015, WRR and many others). Here, please note that the SAS-function approach is functionally and even mathematically (!!) equivalent to the mixing coefficient approach above, as described in detail in Hrachowitz et al. (2016; Wires). The only difference is in the choice of the SAS-function (piece-wise linear for mixing coefficient approach and typically gamma or beta distributions for SAS approach) and in the way the involved processes are described semantically. It is true that many applications(!) of the SASfunction approach limit themselves to overly simplified representations of hydrological systems. However, this does not mean that the general concept behind it is invalid. The application of different individual(!) mixing coefficients (and thus SAS-functions) for different individual(!) system components has, on the contrary, already proven very useful (Fenicia et al., 2010, WRR; Hrachowitz et al. 2013, HESS; Hrachowitz et al. 2015, HP; Hrachowitz et al. 2016, Wires). With the new information provided in this manuscript, a logical next step should thus be to check if these types of models can reproduce power law transit time distributions and if power law distributions would besuitable as SASfunctions in different components of the system (e.g. in the unsaturated root zone for drainage and evaporation; in groundwater for drainage). In fact in our paper Hrachowitz et al. (2013; HESS), using incomplete, temporally varying mixing in the unsaturated zone, we found that the system overall transit time distributions (i.e. TTD of the modelled combined outflows) had longer than gamma distribution tails. An observation that we could not make too much sense of at that time. However, it could fit very well into

**HESSD**
the reasoning presented by the authors here. In that sense, this would be a beautiful example of how different modelling approaches could learn from each other. It would make the manuscript much stronger if the authors emphasized how their findings could be helpful for other modelling approaches and if the authors invested some more effort in being more accurate in their description of what different model approaches can do. In the current description the authors seem to equate what "is" done with these models with what "can" be done. This is not a valid assumption. In fact we can do much more than is typically done.

**RESPONSE:** As we stated above, we very much agree with MH about the complementary merits of both model worlds and will revise the related sections accordingly. There is nothing to argue against integral approaches for simulating transport. At the end of the day, the CTRW approach is also an integral approach. The beauty is that the latter can be linked – we think in a straightforward manner – to distributed models as acknowledged by MH.

Minor comments:

MH: p.3,l.96: what are "uncorrelated noise pattern"?

**RESPONSE:** An uncorrelated noise pattern is a variable field without a spatial correlation. This means that the value at any grid point bears no predictive information about the neighbor. The analogue is a white noise time series, i.e., a purely random signal without temporal autocorrelation.

**MH:** p.4,I.133: Hrachowitz et al. (2016) would fit nicely here as reference. **RESPONSE:** Absolutely, we will include the reference.

**MH:** p.4,I.146: not sure why this is referred to as "weak interaction" here. Given its importance for flow velocities and sediment transport, should friction not cause a "strong interaction" between water and solids?

**RESPONSE:** These terms were introduced in Kleidon et al. (Kleidon et al., 2013). Of course, friction is in both cases, for open channel flow and for porous media flow, the

HESSD
main free energy sink. In an open channel, energy losses occur only at the wetted perimeter, and flow is slowed down in the boundary layer close to the contact line, but there is also a large cross-section where free flow occurs. In a porous medium, friction is everywhere, because the fluid is in contact with the solid at the entire inner surface - so boundary layer effects dominate the entire flow. Note that the maximum acceleration acting on surface water is one g (g dz/dz); in partially saturated soil, it can 50 g (g dïĄź/dz), but flow velocities are orders of magnitude slower, because dissipative losses are larger. This is what we mean by strong interaction. We will include this clarification in the revised manuscript.

**MH:** p.5,I.161 and elsewhere: why only chemicals? This is also true for the behaviour of individual water molecules. The difference is that we can normally not tag and distinguish them. But this does not make the general process valid for chemicals only. There is also imperfect mixing of waters of different ages/provenance. Please try to be more precise in these formulations. In addition, please note that this is exactly what incomplete mixing assumptions and/or non-uniform SAS-functions try to mimic.

**RESPONSE:** Thanks for this point, but here we need to be really precise. We cannot measure diffusion of H2O molecules in H2O, but we can measure diffusion of HDO and HTO in H2O. The velocity of these molecules will indeed differ from the fluid velocity, but it is not for the H2O fluid molecules. In the revised manuscript, we will add a note to clarify this and explain our focus on chemicals. This point is also addressed explicitly in the second paragraph of section 3.1. Additional note: Diffusive mixing of HDO and HTO is, however, a problem with regard to the existence of stable end-members. Assume two water bodies of largely different isotopic signatures, young and old water, in contact. Diffusive mixing among both water bodies will alter the apparent age of the probe to an intermediate age, even if there is no physical mixing.

**MH:** p.5,I.178: also depending on the pre-storm wetness conditions **RESPONSE:** Of course, we will add this.

MH: p.6,I.203-204: I disagree with this statement. While it is true that conceptual mod-
els are often applied in a careless way, there is a wide body of literature that describes the necessity and value of treating models as hypotheses that need to be tested (e.g. Fenicia et al., 2014; Kavetski et al., 2011; Clark et al., 2011;2015), which, in an iterative approach, allows us to learn about the system. In addition, in Hrachowitz and Clark (2017) we argue that ideally, these catchment-scale, effective parameters should reflect real world numbers. The actual challenge is to find these numbers, which is far from trivial for many parameters, but which was shown to be feasible for others, e.g. Master-Recession Curve (Lamb and Beven, 1997) or the storage capacity in the unsaturated root zone (e.g. deBoer-Euser et al., 2016; Nijzink et al., 2016)

**RESPONSE:** We agree that this is a sensible point. Allow us to expand on what we mean by the statement "that conceptual models provide an abstract explanation". "Explaining" means to us to provide reasons that are consistent with the underlying theory and that can be tested independently from the data that have been used to train the model. When explaining the result of a mathematical model, it is of course sufficient to get a mathematical explanation, based on arithmetic or algebra. Hydrological systems are, however, physical systems, which means that we need to provide the physical causes for the observed effects. A "beta" of 9 in conjunction with the HBV beta may represent a mathematically feasible reason, but it is not a physical cause. Furthermore, this explanation cannot be tested by taking independent observations as beta is neither directly observable nor is there a clear, unique relation to observable catchment characteristics. For a reductionist model, we can explain the model performance using, e.g., the fill and spill idea and the related storage volume in bedrock pools. This can be observed using geophysics or augers and bedrock can be represented in the model (see(Loritz et al., 2017)) via the subsurface permeability field. The structure of the model is an image of the structure of the landscape, and depths to bedrock can be inferred from data or related assumptions can be tested in a fashion that is independent of the calibration exercise. To us, this is (by the way) the core idea of PUB - building models that can be informed by data which are independent of discharge.

In the revised manuscript, we will better stress that these points, specifically in terms of
the representation of partially saturated soil in conceptual models. We do not question the value of a master recession analysis nor the concept of dynamic root zone storage. We personally also think that statements like "this area works like a plateau", "slope or wetlands is a cause", etc., are very informative, in cases when landscape entities can represented by unique and typical conceptual model building blocks. We will stress this even more in the revised manuscript, although we are not entirely convinced that conceptual modelling has already reached this stage.

**MH:** p.6,I.210: I also disagree here. These relations remain specific for the spectrum of environments they have been developed for. If they relations were developed using a wide range of different landscapes, as done for example for the determination of global parameters in the mhM model (e.g. Samaniego et al., 2010) than they will also give us a more general picture.

**RESPONSE:** This is indeed a valid point, and we will revise this statement to stress that the difference between the HBV and the mHm is the more realistic representation of the partially saturated soil. The mHm uses the retention curve after Brooks-Corey (Brooks and Corey, 1964) for soil moisture accounting (not the beta store). This is a reasonable representation of capillary forces acting on soil water and the additional asset is that the entire knowledge of the soil physical community can be in the regionalization.

**MH:** p.6,I.219: No, I strongly disagree! Why should a parameter that describes the aggregated effects of heterogeneity be an abstract quantity? I think it rather provides the macroscale perspective and it can in some instances already be independently observed/estimated (see comment above about Master-Recession-Curve and storage capacity in root zone)

**RESPONSE:** As we stated above, we mainly criticized the representation of the partially saturated zone and of hillslope scale processes. Given the success of the mHm it makes a difference whether the beta store concept of a retention function is used. We do not question the Master Recession curve, as it may be nicely related to the Darcy equation (de Rooij, 2013). Also storage root zone capacity is a valuable concept. But HESSD
the latter does not imply that soil is unimportant, as is sometimes stated. There would not be any storage of water without capillarity. So the term "root zone storage" implies that capillarity exits, otherwise the roots have to tap groundwater.

Last but not least, a simple mathematical representation does not automatically mean that the model structures and the parameters are straightforward to interpret. This is evident not only in the fact that internal conceptual model states and parameters are difficult to compare and derive from field observations, but also in that model structure is neither simple nor intuitive to interpret. For example, the FLEXTOPO model structure is abstract, and it is not simple to infer the plateau, slope and wetland from the corresponding combination of storage elements. Physically-based models are not based on simple mathematical equations, but their spatial setup reflects much more intuitively our perception of how the system looks, particularly in the subsurface. And at the end of the day, at least finite differences based numerical solutions are founded on algebra and thus by no means more complex than the math underlying conceptual models.

**MH:** p.6,I. 223-224: no, although there is without doubt some tendency to believe that parameter search can help, I do not share this. I rather think that, for all the uncertainties involved, we need to start with step-by-step limiting the feasible parameter space by identifying and eliminating solutions that are inconsistent with our data and our understanding of the system. That is essentially the opposite of finding the "optimal" parameter set and relatively independent of parameter search algorithms.

**RESPONSE:** Agreed, but as we discuss elsewhere in the manuscript, it is preferable to constrain parameters that have physical meaning (such as root zone storage or saturated hydraulic conductivity).

**MH:** p.6,I.231-234: with all appreciation for the authors, but this is too bold a statement, which I need to consider as plainly wrong. First, these models \*can\* resolve the celerity/ velocity dualism, when implemented as described above (mixing coefficients/SAS function). And this, second, has already been demonstrated with a plethora of articles on combined modelling of water flow and (non-)conservative solute concentrations
(e.g. chloride, nitrate, silica, ANC, EC, DOC and many others; see above references and references therein)

**RESPONSE:** In the context of these specific examples, we agree completely and will revise the manuscript accordingly. As an aside, though, we wonder whether the runoff components simulated with conceptual models can be used to constrain the average particle velocities in the respective flow domains. The latter would require being specific about the wetted cross section in these flow domains, with the mass balance used to partly constrain particle transit time distributions. In this respect, the reductionist model approach may be somewhat more consistent, but this might be seen as a matter of taste.

**MH:** p.7,I.247-263: while Darcy-Richards does indeed, and probably rightfully so, receive criticism, I think these problems can be fixed within the near future (as also suggested by the list of improvements given here by the authors). I rather consider another point why these models receive much criticism: the fact that the sheer number of parameters needed can never be observed at the spatial resolution and scale(!!) of interest. Either we then need to use our scarce, existing observations to inform our model, in which case we will upscale homogeneity (as the spatial correlation fields of our system properties are unknown!), or we will need to calibrate these models, in which case we will obviously run into the problem of equifinality and our inability to meaningfully constrain our models.

**RESPONSE:** We think it is very important to critique Darcy-Richards models, as they rely on incomplete physics, just as it is important to comment on conceptual models. Some of the criticism is simply wrong. Darcy-Richards models can be informed using distributed data sets, at the catchment scale, although this is often claimed to be impossible. Darcy-Richards models can simulate preferential flow as well as conceptual models. We do not see that they contain so many more parameters; the Brooks and Corey model has 4 parameters, two of which are easily constrained, the beta store has 2 parameters, neither of which is easily constrained. We fully agree that they are subject to equifinality, but the parameter sets are also easier to be constraint. At the
end of the day we can use hopefully infer these parameters from thermodynamic optimality. We note that the power law exponent in the breakthrough curve, as analyzed by means of the CTRW concept, has a clear relation to aquifer characteristics (e.g., mean, variance of conductivity) as has been shown by Edery et al. (2014). In the likely case that there is also a clear relation to the correlation length, it may be possible to use breakthrough curves to infer this information and parameterize distributed models. We will add a comment in this vein in the revised manuscript.

**MH:** p.7,l.264-265: why "alternative"? should these tracers not also be very informative and helpful to implement and test the above models? **RESPONSE:** Agreed. We will modify the wording here.

**MH:** p.7,1.264-270: true, but mostly outdated, as these approaches are too simplistic. **RESPONSE:** Given that this is "true", we prefer to retain this material. However, in the revised manuscript, we will note that newer approaches have been advanced.

**MH:** p.7,I.271-283: in this paragraph the authors seem to be confusing different approaches. At least it reads in a rather incoherent way. For example, Rinaldo et al. (2015) and Botter et al. (2011) describe the same general concept. Further, the system overall SASfunction can also emerge from the choice of SAS functions from individual components, which can be a calibration parameter or informed by observations/theoretical considerations as demonstrated by the authors of this manuscript. Finally, yes, the gamma distribution is often used, but as often other choices, such as the beta-distribution (e.g. van der Velde et al., 2012) or some piece-wise linear distributions are made (Hrachowitz et al., 2013,2015). In any case, and as mentioned above, these choices are not necessarily made out of conviction but rather due to a lack of more information and can be easily adjusted. In particular, it will be very help to get a better understanding how the tails of the distributions should look like.

**RESPONSE:** We thank MH for pointing this out and will revise this section accordingly. We agree strongly that the inverse Gamma function might be a useful concept in this context.

HESSD
**MH:** p.10,I.364-370: unclear in how far this is different to what mixing coefficient/SASfunctions do.

**RESPONSE:** We agree with MH that in an integral context, age-ranked storage reflects this continuum of scales and water ages. This part of the discussion focuses on physical processes and dynamics, and how to conceptualize them, not how to model them. So we do not see any need to refer specifically to mixing coefficients (as designed for \*any\* of a variety of specific models) or SAS functions. We will add a note in the revised manuscript.

**MH:** p.10,I.375-378: I believe there is quite a good understanding in the community that there is no binary distinction between separated pools of water. The "two water worlds" need to be understood, prosaic literature terms, as a hyperbole – an exagger-ated analogy, i.e. a pointed description of a concept.

**RESPONSE:** Thanks for pointing this out. While we acknowledge the usefulness of metaphors as a means of communicating science in an appealing way, this should be not overdone. There is evidence that pockets of mobile and immobile water ages can exist in very close spatial proximity; these pockets will have strongly different ages. And it is clear that there will be diffusive exchange of, e.g., stable isotopes between these pockets (as discussed above). This suggests that a distinction of different water worlds based on isotopic signatures is not straightforward. Every distinction that goes beyond the well-established blue and green water concept of Marlin Falkenmark, which is in fact based on straightforward soil physics and can be inferred from the soil water retention curve, remains highly uncertain. Given the extensive debate and literature on this subject, what is a "good understanding" can be considered uncertain. We state clearly that this picture of a distinct separation is indeed a highly idealized interpretation, and include this discussion as part of our overall appraisal of the "critical zone" in section 2.2. However, in light of MH's comment, we will clarify this point further in the revised manuscript.

MH: p.10,I.379-380: perhaps better "drainage and evaporative fluxes"
**RESPONSE:** Agreed. We will modify in the revised manuscript.

**MH:** p.11,I.431 and elsewhere: I am not sure if the term "chemical transport" is the best way to express what you want to express here. why not keeping it more general to "transport", which then indeed is further divided into physical, bio-physical and chemical components in the transport processes.

**RESPONSE:** This term is used often in the groundwater literature. Here, we are referring explicitly and specifically to transport of chemical species, which is of course affected by physical and biogeochemical processes.

MH: p.12,I.454: conservative transport?

**RESPONSE:** Yes, the simulations here assumed that the chemical is conservative. We will clarify this in the revised manuscript.

MH: p.14,I.500: where can I see this?

**RESPONSE:** This inadequacy of the advection-dispersion equation is seen, for example, in Figure 4 (which is referred to in I.495); see the red dashed-dotted curve. We will clarify this in the revised manuscript.

**MH:** p.15,I.500-503: sure, but would be good to introduce what CTRW is before shown results.

**RESPONSE:** This section 2.3 focuses on a more general conceptual discussion, in keeping with the style of sections 2.1 and 2.2. We state clearly that the CTRW framework and governing transport equations are detailed in section 3.3; we prefer to retain the explanation of CTRW in there, which leads naturally into the quantitative analysis of various transition time distributions and long tailing of overall transit times.

**MH:** p.15,I.550ff: as also demonstrated by Hrachowitz et al. (2015 – sorry to again bring in one of my own papers here) using a conceptual model with mixing coefficients, where we showed that conservative (!) chloride transport is slower than water transport (evapoconcentration)

**RESPONSE:** Excellent. We will include here this reference and the result in the revised
manuscript.

**MH:** p.17,I.594 ff: I really like this analogies here, but a stronger explanation of and link between spatial and temporal heterogeneity needed. Why are they different? What is going on there?

**RESPONSE:** We think that the key is to think in space-time and complementary manifestations of heterogeneity of preferential flow. Heterogeneous flow patterns that emerge in space (defined as deviation from uniform flow, which is the maximum entropy pattern), translate into signatures in the breakthrough curve, which are observed at a fixed location as a function of time. Again, this is through deviations from Gaussian travel times (which is the maximum entropy travel time distribution, reflecting wellmixed, advective-dispersive transport). When looking at the example of Edery et al. (2014), it is important to stress that in the preferential flow paths, particles pass through "bottlenecks", which would be excluded as possible flow path when using, for instance, percolation theory. This is because the higher local gradient in pressure head "pushes" the water through the bottleneck. As the gradient in pressure head reflects the potential energy difference, this corresponds to a large flow against a large gradient and thus a local power maximum. We will further emphasize this in the revised manuscript. The overall key messages of section 3 are that (a) CTRW is a consistent with the ADE and advances beyond it, particularly in terms of capturing dispersion and tailing effects, (b) the power law exponent is related to porous media characteristics as well as the flow conditions, so that we can infer spatial signatures from temporal ones. In the revised manuscript, we will also clearly articulate these messages in the introductory discussion.

**MH:** p.17,I. 632ff, section 3.3: in how far is this different to SAS? This does not clearly come out here. I think it would be very interesting for the reader to clarify this. Is there a fundamental difference or is it only the choice of the transition time distributions? **RESPONSE:** We realize that there is lack of clarity regarding "travel time distributions" and "transit time distributions") in the manuscript. A "travel time distribution" refers
to the overall response of a domain – catchment, soil column, or aquifer – from an input point/plane to an output (measurement) point/plane. We will certainly clarify this critical point in the revised manuscript, both when we first introduce these terms and throughout the manuscript, where appropriate. And we agree that this is linked strongly to age-ranked storage, which is the integral of the travel time distribution over all ages.

**MH:** p.17,I.646 and elsewhere: I noted the use of "transition" times instead of what is standard use in surface water hydrology: "transit" times. Was this made on purpose? Is there a difference? If yes, please specify. If no, please also clarify.

**RESPONSE:** Please see our response to the previous comment. We certainly agree that there is a lack of clarity and uneven use of terminology in our communities. We will clarify this key point in the revised manuscript.

**MH:** p.17,I.670: water molecules, too, are subject to both! It is just that its more difficult for us to observe.

**RESPONSE:** We pointed this out explicitly in the discussion in the second paragraph of section 3.1. In the revised manuscript, we will refer to that discussion here, too, for clarity.

**MH:** p.21,I.802-804: because that was the best information we had so far. **RESPONSE:** Agreed. We were not criticizing anyone here, only stating a fact.

**MH:** p.22,I.821: not sure this is correct ". . .sum of a gamma function (which is also a gamma function). . .". Please check if this makes sense. Do you mean the scaled incomplete gamma function? Should it not rather reader something like:"the sum of 2 random variables that follow a gamma distribution can also be described by a gamma distribution"?

**RESPONSE:** Thank you for noting this. We certainly agree (our wording was poor), and we will correct this in the revised manuscript.

**MH:** p.22,I.824: well, no. Not if suitable local (i.e. individual for each system component) mixing coefficients/SAS-functions are chosen.

HESSD
**RESPONSE:** We agree that the proper choice of an SAS goes beyond the assumption of perfect mixing. We will revise this part. The key point is, however, that we have been using similar concepts to characterize routing and transport; both phenomena are affected by preferential flow and we use the same distribution function to describe the related fingerprints of preferential flow. We think the CTRW concept is somewhat more general in capturing both the short tail and the long tails. We will stress this even more.

**MH:** p.22,I.839-841: I am not sure that that reference is a good example. First, the NSE is a very pessimistic performance metric in cases there is a low signal-to-noise ratio in the target variable (which is the fact in the Weiherbach: the stream isotope composition is strongly damped and essentially plots close to a straight line – any small deviation from that – error or real process – causes strong effects on NSE). Second, the choice of model also only allows a rather rudimentary partitioning and routing of water fluxes, which will have a considerable effect on the tracer composition.

**RESPONSE:** We agree and we will revise this section accordingly. The overall key messages of section 3 are that (a) CTRW is a consistent with the ADE and advances beyond it, particularly in terms of capturing dispersion and tailing effects, (b) the power law exponent is related to porous media characteristics as well as the flow conditions, so that we can infer spatial signatures from temporal ones. We are pleased that we can make these points without criticizing the work of Rodriguez et al. (2019).

**MH:** p.23,I.868-873: it is difficult to judge for the reader if the CTRW framework is more physically justified than other models. What also remains unclear: is CTRW necessary to model long tails? Or can suitable distributions be used in other (conceptual) models to reproduce similar results?

**RESPONSE:** First, the CTRW framework is physically justified, and certainly is highly effective in capturing long tails. The references to CTRW cited throughout this manuscript provide the background (and, e.g., Figure 4 demonstrates long tailing phenomena); clearly, we cannot review all of the evidence for this in the current manuscript.
Second, we are not claiming that CTRW is more physically justified than other \*physical\* models; we state only that CTRW is physically justified, as is incorporation of an inverse Gamma transition time distribution. The reviewer agrees, for example, that use of a Gamma distribution is convenient, but has not been given a physical basis for its use. Finally, we do not claim that other suitable distributions are unable to reproduce similar results, although we are unaware of any other \*physically-justified\* model that yields such results. In the revised manuscript, we will include further text clarifications on each of these points.

References: Brooks, R. H., and Corey, A. T.: Hydraulic properties of porous media, Hydrology Paper, 3, 22-27, 1964. de Rooij, G. H.: Aquifer-scale flow equations as generalized linear reservoir models for strip and circular aquifers: Links between the Darcian and the aquifer scale, Water Resour. Res., 49, 8605-8615, doi:10.1002/2013WR014873, 2013. Klaus, J., and Zehe, E.: Modelling rapid flow response of a tile drained field site using a 2d-physically based model: Assessment of "equifinal" model setups, Hydrological Processes, 24, 1595 - 1609, DOI: 10.1002/hyp.7687., 2010. Klaus, J., and Zehe, E.: A novel explicit approach to model bromide and pesticide transport in connected soil structures, Hydrology And Earth System Sciences, 15, 2127-2144, 10.5194/hess-15-2127-2011, 2011. Kleidon, A., Zehe, E., Ehret, U., and Scherer, U.: Thermodynamics, maximum power, and the dynamics of preferential river flow structures at the continental scale, Hydrology And Earth System Sciences, 17, 225-251, 10.5194/hess-17-225-2013, 2013. Loritz, R., Hassler, S. K., Jackisch, C., Allroggen, N., van Schaik, L., Wienhöfer, J., and Zehe, E.: Picturing and modeling catchments by representative hillslopes, Hydrol. Earth Syst. Sci., 21, 1225-1249, 10.5194/hess-21-1225-2017, 2017. Wienhoefer, J., and Zehe, E.: Predicting subsurface stormflow response of a forested hillslope - the role of connected flow paths, Hydrology And Earth System Sciences, 18, 121-138, 10.5194/hess-18-121-2014, 2014.

533, 2019.

---

## Short Comment (SC1) · 4 Dec 2019

I thank Brian Berkowitz and Erwin Zehe for outlining their views on heterogeneous flow and transport of water and solutes in the subsurface. Their manuscript offers several interesting aspects and I especially like the emphasis of connecting different communities. However, I would like to ask Berkowitz and Zehe to reconsider their use of "Two Water Worlds" throughout the manuscript.

I highly recommend to not use this term for the following reason:

1.   The term was introduced by McDonnell (2014) as "vegetation and streams returning different pools of water to the hydrosphere". The term is quite exclusively

used in this very specific context – mostly in isotope hydrology (see list of citing literature here: https://scholar.google.de/scholar?cites=9670915851738538320&as_sdt=5,34&sciodt=0,34&hl=de). Contrary to your introduction, I am not aware that the groundwater community uses the term "Two Water Worlds". I do not see that groundwater hydrologists address the hypothesis posed by McDonnell nor do they use that term in a different way. (see: https://lmgtfy.com/?q=%22Two+Water+Worlds%22)

2. Based on the definition from McDonnell, the term "water world" is not correctly used in this manuscript, when the authors state for example in L43: "...two systems – surface water and groundwater – using the (often distinct) terminology of each of these "water world" research communities." The "Two Water Worlds" are not surface water vs. groundwater.

3. It is not correct that the term "Two Water Worlds" was used by Brooks et al. (2010) as you state in L576. Brooks et al. introduced "ecohydrological separation".

4. Since it is stated in L579 that "We question the conceptualization of two (or more) separate, fully compartmentalized mobile and immobile regions of water and chemicals.", why would one continue using the term "Two Water World"? Why promoting an oversimplified expression about which you acknowledge in your response to Markus Hrachowitz (page C15) that a "distinct separation is indeed a highly idealized interpretation"?

5. The "Two" in "Two Water World" resulted from the two different methods to sample the isotopic composition (2H and 18O) of subsurface water (as done in the early work on "ecohydrological separation" by Brooks et al. (2010) and Goldsmith et al. (2012)): One is either limited to the more "mobile soil" water when using suction lysimeters (often about 600 hPa) or one samples the entire pore water ("bulk soil water"; i.e., mobile and more tightly bound water) by using for example cryogenic extraction. I discussed these aspects in more detail in Sprenger et al. (2018) and Sprenger et al. (2019). Thus, the limitation to TWO separate subsurface pools is to a great extend a result of the methodological limitations, since we cannot simply sample stable isotopes along the water retention curve (but some attempts were done, see e.g. Figure 4 in

[Figure]

Geris et al. (2015)).

I am concerned that the use of "Two Water Worlds" in this manuscript will cause confusion among the hydrological community and I hope that the points I raised here will encourage the authors to use a different terminology.

References

Brooks, J. R., Barnard, H. R., Coulombe, R., and McDonnell, J. J.: Ecohydrologic separation of water between trees and streams in a Mediterranean climate, Nat. Geosci., 3, 100–104, doi:10.1038/NGEO722, 2010.

Geris, J., Tetzlaff, D., McDonnell, J., Anderson, J., Paton, G., and Soulsby, C.: Ecohydrological separation in wet, low energy northern environments? A preliminary assessment using different soil water extraction techniques, Hydrol. Process., 29, 5139–5152, doi:10.1002/hyp.10603, 2015.

Goldsmith, G. R., Muñoz-Villers, L. E., Holwerda, F., McDonnell, J. J., Asbjornsen, H., and Dawson, T. E.: Stable isotopes reveal linkages among ecohydrological processes in a seasonally dry tropical montane cloud forest, Ecohydrol., 5, 779–790, doi:10.1002/eco.268, 2012.

McDonnell, J. J.: The two water worlds hypothesis: ecohydrological separation of water between streams and trees?, WIREs Water, 1, 323–329, doi:10.1002/wat2.1027, 2014.

Sprenger, M., Llorens, P., Cayuela, C., Gallart, F., and Latron, J.: Mechanisms of consistently disjunct soil water pools over (pore) space and time, Hydrol. Earth Syst. Sci., 23, 2751–2762, doi:10.5194/hess-23-2751-2019, 2019.

Sprenger, M., Tetzlaff, D., Buttle, J. M., Laudon, H., Leistert, H., Mitchell, C. P. J., Snelgrove, J., Weiler, M., and Soulsby, C.: Measuring and modelling stable isotopes of mobile and bulk soil water, Vadose Zone J., 17, 170149, doi:10.2136/VZJ2017.08.0149, 2018.

[Figure]

533, 2019.

---

## Author Comment (AC2) · 6 Dec 2019

We sincerely thank Matthias Sprenger (MS) for his thoughtful and constructive comments on our manuscript. We respond below to the individual points.

**MS:** I thank Brian Berkowitz and Erwin Zehe for outlining their views on heterogeneous flow and transport of water and solutes in the subsurface. Their manuscript offers several interesting aspects and I especially like the emphasis of connecting different communities. However, I would like to ask Berkowitz and Zehe to reconsider their use of "Two Water Worlds" throughout the manuscript. I highly recommend to not use this term for the following reason:

[Figure]

**RESPONSE:** We are pleased MS finds that the manuscript offers interesting aspects and connects different communities. While we are fully cognizant that our use of the term "Two Water Worlds" is new and possibly somewhat unusual, we prefer to retain it. Throughout the manuscript, we emphasize the need for connection and cross-fertilization of concepts and methods between the surface water (catchment hydrology) and groundwater communities, which are currently split into two "water worlds". We state this clearly at the outset of the manuscript, and elsewhere. However, throughout the revised manuscript, we will ensure that we are clear on use of this term.

**MS:** 1. The term was introduced by McDonnell (2014) as "vegetation and streams returning different pools of water to the hydrosphere". The term is quite exclusively used in this very specific context – mostly in isotope hydrology (see list of citing literature here: https: ....). Contrary to your introduction, I am not aware that the groundwater community uses the term "Two Water Worlds". I do not see that groundwater hydrologists address the hypothesis posed by McDonnell nor do they use that term in a different way. (see: https: ...)

**RESPONSE:** While McDonnell use the term in a specific context, as noted by MS, there is no exclusivity or "monopoly" in the use of the term. Isotope hydrologists may indeed prefer this term to refer to vegetation and stream pools. In this respect, we note in the manuscript that it does not necessarily properly describe the actual physical situation and dynamics. Thus the term "Two Water Worlds" is a metaphor to better highlight the fact that plants may tap storage fractions in the subsurface which do not contribute to streamflow generation. We do not claim in the manuscript, and certainly do not mean to imply, that the groundwater community uses the term "Two Water Worlds" in the sense it has been introduced by McDonnell (2014). Rather, we introduce the term here to refer to the two worlds again in a metaphoric sense to better illustrate the largely disjunctive nature of the catchment hydrology/surface water and groundwater communities.

**MS:** 2. Based on the definition from McDonnell, the term "water world" is not correctly

used in this manuscript, when the authors state for example in L43: ". . .two systems – surface water and groundwater – using the (often distinct) terminology of each of these "water world" research communities." The "Two Water Worlds" are not surface water vs. groundwater.

**RESPONSE:** As noted above, we introduce the term "Two Water Worlds" to emphasize the current separation between the two worlds – communities – of surface water and groundwater. With all due respect for his excellent research, McDonnell does not hold exclusivity over this term, which we in fact find to be somewhat misleading in terms of description of the true physical picture.

**MS:** 3. It is not correct that the term "Two Water Worlds" was used by Brooks et al. (2010) as you state in L576. Brooks et al. introduced "ecohydrological separation".
**RESPONSE:** We thank MS very much for pointing this out. We will correct the wording and citation to this term (McDonnell, 2014) in the revised manuscript.

**MS:** 4. Since it is stated in L579 that "We question the conceptualization of two (or more) separate, fully compartmentalized mobile and immobile regions of water and chemicals.", why would one continue using the term "Two Water World"? Why promoting an oversimplified expression about which you acknowledge in your response to Markus Hrachowitz (page C15) that a "distinct separation is indeed a highly idealized interpretation"?
**RESPONSE:** By questioning the conceptualization of two separate compartments, we explicitly do not recommend further use of the term "Two Water Worlds", after McDonnell (2014). Allow us to point out again that we use this metaphor in a totally different context, to emphasize the current separation between the two hydrology communities of surface water and groundwater.

**MS:** 5. The "Two" in "Two Water World" resulted from the two different methods to sample the isotopic composition (2H and 18O) of subsurface water (as done in the early work on "ecohydrological separation" by Brooks et al. (2010) and Goldsmith et al. (2012)): One is either limited to the more "mobile soil" water when using suction

lysimeters (often about 600 hPa) or one samples the entire pore water ("bulk soil water"; i.e., mobile and more tightly bound water) by using for example cryogenic extraction. I discussed these aspects in more detail in Sprenger et al. (2018) and Sprenger et al. (2019). Thus, the limitation to TWO separate subsurface pools is to a great extend a result of the methodological limitations, since we cannot simply sample stable isotopes along the water retention curve (but some attempts were done, see e.g. Figure 4 in Geris et al. (2015)).

**RESPONSE:** We sincerely thank MH for pointing this out, and we will happily clarify our text in the revised manuscript, adding these useful citations.

**MS:** I am concerned that the use of "Two Water Worlds" in this manuscript will cause confusion among the hydrological community and I hope that the points I raised here will encourage the authors to use a different terminology.

**RESPONSE:** While we are fully cognizant that our use of the term "Two Water Worlds" is new and possibly somewhat unusual, we prefer to retain it. Throughout the manuscript, we emphasize the need for connection and cross-fertilization of concepts and methods between the surface water (catchment hydrology) and groundwater communities, which are currently split into two "water worlds". We state this clearly at the outset of the manuscript, and elsewhere. We will ensure that we are clear on this point throughout the revised manuscript. We do not believe that use of the term will cause any real confusion. Rather, we are of the opinion that using the term helps to stress the current – but unfortunate and undesirable – lack of communication between the surface water and groundwater worlds.

---

## Referee Comment (RC2) · Anonymous Referee #2 · 10 Dec 2019

The authors present a contribution stressing the observation that surface and subsurface systems should be described by a single model, since both systems are "a manifestation of self-organization". While (of course) I agree with the obvious observation that surface and subsurface systems are governed by the same physical principles (conservation of mass, momentum and energy) I do not agree (in general) with the observation that a "single model" can efficiently and with the same level of accuracy capture the behavior of all systems. Taken to the extreme: we all know that the Navier-Stokes (NS) equations can describe incompressible fluid flow in (simple and complex) systems. However, direct solution of the NS equations is typically not feasible (in general) for turbulent flow or flow in large aquifers. This is why, in various disciplines and

with reference to specific topics, diverse (simplified) models/approaches have been developed, both with reference to surface and subsurface flow conditions.

Then, the authors stress the ability of Continuous Time Random Walk (CTRW) to describe non-Fickian transport in heterogeneous (surface and subsurface) systems. CTRW has a long history (it was originally introduced by Montroll and Weiss (1965), to the best of my knowledge). It has been widely used by the subsurface hydrology community and it allows including the impact of heterogeneity on transport. The model fully depends on the pdf of transition time (i.e., the weighting time in between two jumps). This pdf is an input function in CTRW; it has to be known "a priori" and it is (usually) modeled as a truncated power law, thus embedding fitting parameters. Here, the authors argue that CTRW could be used also to simulate transport in surface systems. Indeed, several studies along this line have already been presented in the literature, as also acknowledged by the authors, albeit not at the catchment- scale.

In summary, I do not clearly see the novelty of the present contribution. No original works/results are presented. The manuscript looks like an "opinion" paper where the authors present an overview of previous work in surface and subsurface systems (with particular emphasis to CTRW approach). The "novelty" should be the suggestion of future works where CTRW could be applied to simulate the transport feature at the catchment scale. This observation appears not at all surprising to me. CTRW is a tools allowing to embed the effect of the heterogeneity of the system on transport features via the use the transition time distribution, regardless the system considered (surface of subsurface). The drawback of this approach is that the pdf of the transition time must be known "a priori" as well as its parameters (that are fitting parameters and must be estimated via available data).

In conclusion, given the flavor of the study (at least the way it is perceived through my analysis), my suggestion would be to reconsider the scope of this contribution. This can be achieved it by framing it in the context of a review or, probably better, an opinion paper. This is the spirit with which I would recommend a set of major revisions.

---

## Author Comment (AC3) · 14 Dec 2019

We sincerely thank Anonymous Referee 2 (AR2) for his/her comments on our manuscript. We respond below to the individual points.

**AR2:** The authors present a contribution stressing the observation that surface and subsurface systems should be described by a single model, since both systems are "a manifestation of self-organization". While (of course) I agree with the obvious observation that surface and subsurface systems are governed by the same physical principles (conservation of mass, momentum and energy) I do not agree (in general) with the observation that a "single model" can efficiently and with the same level of accuracy

capture the behavior of all systems. Taken to the extreme: we all know that the Navier-Stokes (NS) equations can describe incompressible fluid flow in (simple and complex) systems. However, direct solution of the NS equations is typically not feasible (in general) for turbulent flow or flow in large aquifers. This is why, in various disciplines and with reference to specific topics, diverse (simplified) models/approaches have been developed, both with reference to surface and subsurface flow conditions.

**RESPONSE:** We respectfully disagree with the principal claim in this comment, namely the referee's assessment that we stress "surface and subsurface systems should be described by a single model". We make no such argument anywhere in the manuscript. Predictive modelling of a hydrological system requires essentially solving the mass, energy, momentum and entropy balance equations. We agree with the reviewer that one might use, e.g., approximations of the Navier-Stokes equations for the momentum balance in turbulent open channel or overland flow that are different from those for porous media flow. Indeed, at the outset (second paragraph of section 2), we pointed out that surface flow velocity is proportional to the square root of the potential energy difference, while it is proportional to the head/potential energy difference in the subsurface. In the revised manuscript, we will further expand the text to consider this point more fully.

Hydrological modelling (and hydrological theory) attempts to predict how processes described by equations evolve in and interact with a structured heterogeneous domain (i.e., hydrological landscape). Thus, our key observation that both systems are a manifestation of self-organization does not imply proposed use of a single model. Rather, we argue that similar conceptualizations and methods of quantifications – whether related to preferential flow paths, dynamics and patterning of chemical transport and reactivity, and characterization in terms of energy dissipation and entropy, for example – can and should be applied to both surface and subsurface systems, to the benefit of both research communities. In the revised manuscript, we will ensure that this point is clear throughout the text, particularly in the introduction and concluding sections.

**AR2:** Then, the authors stress the ability of Continuous Time Random Walk (CTRW)

to describe non-Fickian transport in heterogeneous (surface and subsurface) systems. CTRW has a long history (it was originally introduced by Montroll and Weiss (1965), to the best of my knowledge). It has been widely used by the subsurface hydrology community and it allows including the impact of heterogeneity on transport. The model fully depends on the pdf of transition time (i.e., the weighting time in between two jumps). This pdf is an input function in CTRW; it has to be known "a priori" and it is (usually) modeled as a truncated power law, thus embedding fitting parameters. Here, the authors argue that CTRW could be used also to simulate transport in surface systems. Indeed, several studies along this line have already been presented in the literature, as also acknowledged by the authors, albeit not at the catchment- scale.

**RESPONSE:** With regard to our discussion of the CTRW — First, the history of CTRW is reviewed in detail by Berkowitz et al. (Reviews of Geophysics, 2006), as cited in the manuscript; since the referee comments on this, we state here that a random walk with continuous time was introduced by Montroll and Weiss (1965), but the generalization of the formalism to a joint pdf in space and time, labeled "CTRW", and with the physical application to transport, was first given by Scher and Lax, 1973). As noted by the referee, CTRW (introduced to the field of hydrology by Berkowitz and Scher, 1995, 1997, 1998) is now used widely by the subsurface hydrology community, as it accounts well for the impact of heterogeneity on transport. The referee is correct in stating that the governing pdf is an input function in CTRW, which requires fitting parameters. We do not see this as a "criticism", given that every model requires fitting parameters. Ideally, these parameters can be determined from various types of a priori information; indeed, such analyses are incorporated in many types of modelling studies, and they have also been implemented to determine the relevant CTRW parameters.

The referee then continues by (correctly) noting our argument that CTRW could be used also to simulate transport in surface systems. We agree (and cite relevant literature) that several studies along this line have already been presented in the literature, albeit not at the catchment scale. Again, we do not see this as a "criticism" of the manuscript.

If the referee intends that these comments are an indication of lack of novelty, we address this concern in the next point.

**AR2:** In summary, I do not clearly see the novelty of the present contribution. No original works/results are presented. The manuscript looks like an "opinion" paper where the authors present an overview of previous work in surface and subsurface systems (with particular emphasis to CTRW approach). The "novelty" should be the suggestion of future works where CTRW could be applied to simulate the transport feature at the catchment scale. This observation appears not at all surprising to me. CTRW is a tools allowing to embed the effect of the heterogeneity of the system on transport features via the use the transition time distribution, regardless the system considered (surface of subsurface). The drawback of this approach is that the pdf of the transition time must be known "a priori" as well as its parameters (that are fitting parameters and must be estimated via available data).

In conclusion, given the flavor of the study (at least the way it is perceived through my analysis), my suggestion would be to reconsider the scope of this contribution. This can be achieved it by framing it in the context of a review or, probably better, an opinion paper. This is the spirit with which I would recommend a set of major revisions.

**RESPONSE:** We (the two co-authors) discussed this point repeatedly – "opinion paper" or "research paper" – and concluded that while this evaluation is somewhat subjective, the manuscript is best characterized as a research paper. We are gratified, too, that referee 1 and the author of the "short comment" were evidently satisfied with classification of the manuscript as a "research paper". Throughout the manuscript, we synthesize literature, conceptualizations and understanding from the generally separate surface water and groundwater communities. This is much more than an "opinion" or straightforward "review". In doing so, we provide new insights of "commonalities" in the two water worlds, in terms of preferential paths, entropy, and energy dissipation. In terms of the discussion of CTRW, the manuscript offers new analysis and insight showing that one definition of the CTRW power law form of the pdf is actually an inverse Gamma distribution, which meshes beautifully with previous work showing use of an

inverse Gamma rather than a Gamma distribution to characterize transition times and the resulting travel time distributions of chemical tracers. (We address the comment regarding "fitting parameters" in the previous "Comment/Response", and do not repeat here.) Moreover, in terms of the CTRW framework, we stress that the work of Edery et al. (2014) clearly reveals a link between the power law exponent and the underlying medium characteristics. This implies that the fitted parameters are a macroscale fingerprint of spatial media characteristics that determine the temporal arrival of chemical species. While we do not expect that this relation is unique, it does imply that "fitted" parameters have a physical meaning that can be used to constrain characteristics of the domain (i.e., the hydrological landscape mentioned above) in a spatially distributed model. This goes clearly beyond the mere fitting of meaningless/empirical parameters. In the revised manuscript, we will also include a specific (and unpublished) entropy calculation to quantify the preferential paths that control tracer migration. Thus, while our manuscript is not a "traditional" research article ("Methods and Materials, Results, Discussion"), it (i) synthesizes an array of diverse methods for quantification, (ii) offers new insights, (iii) presents new quantitative analysis, and (iv) sets a framework for future research.

---

## Author Response (AR1)

Dear Hubert,

The revision to our manuscript addresses all of the points as described in detail in our point-by-point responses to the reviewers. We therefore refer to those comments and responses, and do not reproduce them here.

The "marked changes" version of the revised manuscript clearly indicates where/what revisions were introduced. However, in addition to those responses, we note the following:

(1) In light of the review comments and further consideration on our part, we decided that the old structure that described and explained concepts along "hydrological compartments" was not ideal. We feel it is better to structure the discussion along the different communities: catchment hydrology and groundwater hydrology. As a consequence, we completely reworked section 2.1 and 2.2 – retaining much of the original text and references in modified form – and added equations, restructured and focused the narrative, and removed as much as possible any "offensive" passages. We therefore did not highlight individual changes in these sections, but instead highlight the entire text.

(2) Sections 4.1, 4.2: note that the original Section 4.2 was deleted in its entirety (too much material for this paper!), and some of the material from section 4.1 was moved to the Conclusions and Perspectives (originally section 5.1). The Conclusions and Perspectives section, formerly section 5, is now section 4.

We hope you will agree that we have fully and thoroughly revised the manuscript to convincingly address the reviewers' comments, as well as yours.

Thank you for your consideration.

Brian and Erwin

[revised manuscript text omitted]

random variables, functions (which is also a gamma distributed function) to characterise
travel time distributions thus implies a conceptualization of the system, in one or several
parallel cascades, of linear reservoirs to representing different flow paths. This in turn rests on
the assumption of locally well-mixed conditions within each reservoir, which is questionable
in the case of preferential flow, and neglects exchange among these cascades along the flow
path. We argue that this is conceptualisation is too idealised, because it does not characterise
the full evolution of ubiquitous power law behaviours (transit/arrival times) over long spatial
and temporal scales. We suggest that the use of an inverse gamma function (or truncated
power law) might improve simulations achieved within past attempts using the gamma
function, reported by, e.g., Harman et al. (2015) and Rodriguez et al. (2019, submitted). The
former study used a single constant gamma distribution for the StorAge Selection (SAS)

[revised manuscript text omitted]

Fenicia, F., Kavetski, D., Savenije, H. H. G., Clark, M. P., Schoups, G., Pfister, L., and Freer, J.: Catchment properties, function, and conceptual model representation: Is there a correspondence?, Hydrol. Proc., 28, 2451-2467, doi:10.1002/hyp.9726, 2014.

Fenicia, F., Savenije, H. H. G., Matgen, P., and Pfister, L.: A comparison of alternative multiobjective calibration strategies for hydrological modelling. Water Resour. Res., 43, W03434, doi:10.1029/2006WR005098, 2007.

Feyen, L., Vazquez, R., Christiaens, K., Sels, O., and Feyen, J.: Application of a distributed physically-based hydrological model to a medium size catchment, Hydrol. Earth Syst. Sci., 4, 47-63, doi:10.5194/hess-4-47-2000, 2000.

Flury, M.: Experimental evidence of transport of pesticides through field soils - a review, J. Environ. Qual., 25, 25-45, doi:10.2134/jeq1996.00472425002500010005x, 1996.

Flury, M., Flühler, H., Leuenberger, J., and Jury, W. A.: Susceptibility of soils to preferential flow of water: A field study, Water Resour. Res., 30, 1945-1954, doi:10.1029/94WR00871, 1994.

Flury, M., Leuenberger, J., Studer, B., and Flühler, H.: Transport of anions and herbicides in a loamy and a sandy soil. , Water Resour. Res., 31, 823-835, doi:10.1029/94WR02852, 1995.

Freeze, R. A. and Harlan, R. L.: Blueprint for a physically-based, digitally simulated hydrologic response model, J. Hydrol., 9, 237-258, doi:10.1016/0022-1694(69)90020-1, 1969.

Gao, H., Hrachowitz, M., Fenicia, F., Gharari, S., and Savenije, H. H. G.: Testing the realism of a topography-driven model (flex-topo) in the nested catchments of the Upper Heihe, China, Hydrol. Earth Syst. Sci., 18, 1895-1915, doi:10.5194/hess-18-1895-2014, 2014.

Gärdenäs, A. I., Šimunek, J., Jarvis, N., and van Genuchten, M. T.: Two-dimensional modelling of preferential water flow and pesticide transport from a tile-drained field, J. Hydrol., 329, 647-660, doi:10.1016/j.jhydrol.2006.03.021, 2006.

Gassmann, M., Stamm, C., Olsson, O., Lange, J., Kümmerer, K., and Weiler, M.: Model-based estimation of pesticides and transformation products and their export pathways in a headwater catchment, Hydrol. Earth Syst. Sci., 17, 5213-5228, doi:10.5194/hess-17-5213-2013, 2013.

[revised manuscript text omitted]

Westhoff, M., Zehe, E., Archambeau, P., and Dewals, B.: Does the Budyko curve reflect a
maximum-power state of hydrological systems? A backward analysis, Hydrol. Earth Syst.
Sci., 20, 479-486, doi:10.5194/hess-20-479-2016, 2016.
Wienhoefer, J. and Zehe, E.: Predicting subsurface stormflow response of a forested hillslope
- the role of connected flow paths, Hydrol. Earth Syst. Sci., 18, 121-138, doi:10.5194/hess-
18-121-2014, 2014.
Wienhofer, J., Germer, K., Lindenmaier, F., Farber, A., and Zehe, E.: Applied tracers for the
observation of subsurface stormflow at the hillslope scale, Hydrol. Earth Syst. Sci., 13,
1145-1161, doi:10.5194/hess-13-1145-2009, 2009.
Wilcke, W., Yasin, S., Valarezo, C., and Zech, W.: Change in water quality during the
passage through a tropical montane rain forest in Ecuador. Biogeochemistry 55, 45-72,
doi:10.1023/A:1010631407270, 2001.
Worthington, S. R. H. and Ford D. C.: Self-organised permeability in carbonate aquifers,
Groundwater, 47(3), 326-336, doi:10.1111/j.1745-6584.2009.00551.x, 2009.
Wrede, S., Fenicia, F., Martinez-Carreras, N., Juilleret, J., Hissler, C., Krein, A., Savenije, H.
H. G., Uhlenbrook, S., Kavetski, D., and Pfister, L.: Towards more systematic perceptual
model development: A case study using 3 Luxembourgish catchments, Hydrol. Proc., 29,
2731-2750, doi:10.1002/hyp.10393, 2015.
Zehe, E. and Jackisch, C.: A Lagrangian model for soil water dynamics during rainfall-driven
conditions, Hydrol. Earth Syst. Sci., 20, 3511-3526, doi:10.5194/hess-20-3511-2016, 2016.
Zehe, E., Blume, T., and Bloschl, G.: The principle of 'maximum energy dissipation': A novel
thermodynamic perspective on rapid water flow in connected soil structures, Philos. Trans.
R. Soc. B-Biol. Sci., 365, 1377-1386, doi:10.1098/rstb.2009.0308, 2010.
Zehe, E., Ehret, U., Blume, T., Kleidon, A., Scherer, U., and Westhoff, M.: A thermodynamic
approach to link self-organization, preferential flow and rainfall-runoff behaviour, Hydrol.
Earth Syst. Sci., 17, 4297-4322, doi:10.5194/hess-17-4297-2013, 2013.
Zehe, E., Ehret, U., Pfister, L., Blume, T., Schroder, B., Westhoff, M., Jackisch, C.,
Schymanski, S. J., Weiler, M., Schulz, K., Allroggen, N., Tronicke, J., van Schaik, L.,
Dietrich, P., Scherer, U., Eccard, J., Wulfmeyer, V., and Kleidon, A.: Hess opinions: From
response units to functional units: A thermodynamic reinterpretation of the hru concept to
link spatial organization and functioning of intermediate scale catchments, Hydrol. Earth
Syst. Sci., 18, 4635-4655, doi:10.5194/hess-18-4635-2014, 2014.
Zehe, E., Loritz, R., Jackisch, C., Westhoff, M., Kleidon, A., Blume, T., Hassler, S. K., and
Savenije, H. H.: Energy states of soil water – a thermodynamic perspective on soil water
dynamics and storage-controlled streamflow generation in different landscapes, Hydrol.
Earth Syst. Sci., 23, 971-987, doi:10.5194/hess-23-971-2019, 2019.
Zhang, Y., Benson, D. A., and Reeves, D. M.: Time and space nonlocalities underlying
fractional-derivative models: Distinction and review of field applications, Adv. Water
Resour., 32, 561-581, doi:10.1016/j.advwatres.2009.01.008, 2009.

---

## Author Response (AR2)

Dear Editor,

We appreciate the careful reading and particularly positive review comments.

In this final version of the manuscript, we have included the three references suggested by the reviewer together with a short description of this research (lines 351-353), and corrected the typo on line 894. We also updated details on several references (papers that were submitted have now been published).

Thank you,
Brian and Erwin